# Probabilistically-routed Bayesian Additive Spanning Trees for Learning on Constrained Domains

Abhisek Chakraborty [1]    Abhishek Mandal [2]    Anirban Chakraborty [3]

## Abstract

Bayesian additive spanning tree (BAST) is an useful tool for interpretable, non-parametric regression on complex constrained domains. It improves upon the performance of Bayesian additive regression trees (BART) by replacing axis-aligned splits through binary tree components by cuts on a spanning tree components, enabling the formation of contiguous splits that respect the underlying complex structure. While BAST is effective for learning on constrained spaces, it still relies on hard partitions, albeit on spanning trees, which limits its ability to represent smoothly varying functions on constrained domains. We propose Probabilistically-routed Bayesian additive spanning trees (PR-BAST), a principled relaxation that replaces hard cuts on spanning tree components with probabilistic routing along spanning tree components. PR-BAST represents the regression surface as an additive ensemble of such spanning tree-aligned smooth components. Conditional on a fixed spanning tree, each component in PR-BAST induces a Gaussian random field with a sparse, tree-structured precision matrix, enabling scalable posterior computation via sparse linear algebra. We theoretically establish that PR-BAST yields strictly faster posterior contraction rates compared to BAST under graph-smooth truth. Experiments on synthetic and real datasets demonstrate that PR-BAST consistently improves accuracy over BAST and other baselines, while retaining the interpretability of tree-based models.

[1]Global Statistical Sciences, Eli Lilly and Company, Indianapolis, United States (Formerly at Department of Statistics, Texas A&M University, College Station, United States) [2]Department of Statistics, Florida State University, Florida, United States [3]Public Health Sciences, Medical University of South Carolina, Charleston, United States. Correspondence to: Abhisek Chakraborty <zovial-papai@gmail.com>.

*Proceedings of the 43$^{rd}$ International Conference on Machine Learning*, Seoul, South Korea. PMLR 306, 2026. Copyright 2026 by the author(s).

## 1. Introduction

We consider non-parametric regression problems where the predictors lie on a constrained complex domain. Observations are modeled as

$$y = f(x) + \varepsilon, \quad \varepsilon \sim \mathrm{N}(0, \sigma^2), \tag{1}$$

where $f$ denotes an unknown function defined over the domain of interest and $\sigma^2$ is an unknown noise variance. In many applications, the function exhibits structured heterogeneity: it varies smoothly along intrinsic connectivity patterns while undergoing abrupt and localized changes driven by domain-specific constraints. A canonical example arises in traffic-flow modeling on road networks with one-way streets, bottlenecks, and flyovers. Traffic intensity typically evolves gradually along connected road segments but may change sharply near capacity-constrained intersections or across elevated flyovers that create long-range connections not captured by Euclidean proximity. Such patterns are naturally aligned with the graph geometry of the road network rather than with ambient spatial coordinates. Capturing this combination of smooth propagation along connected paths, and sharp and abrupt changes when connectivity is disrupted, requires models that adapt to domain geometry while allowing continuous transitions between regions.

Classical tools for non-parametric regression include Gaussian processes (Rasmussen & Williams, 2006; van der Vaart & van Zanten, 2008), spline- (Wahba, 1990) and wavelet-based (Daubechies, 1992; Donoho & Johnstone, 1995) methods, Laplacian smoothing and Gaussian random field models (Zhu et al., 2003; Smola & Kondor, 2003), to name a few. However, tree-based models (Chipman et al., 1998; Breiman, 2001; Friedman, 2001; Chipman et al., 2010; Tan & Roy, 2019; Luo et al., 2021b) are becoming increasingly popular in this setting, as they yield interpretable, localized representations of structural heterogeneity while naturally respecting connectivity constraints, making them well suited for traffic management, and urban infrastructure planning.

Bayesian Additive Regression Trees (BART; Chipman et al. (2010); Linero & Yang (2018); Tan & Roy (2019); Starling et al. (2020); Chipman et al. (2022); Deshpande (2025); Deshpande et al. (2026)) model regression surfaces using ensembles of axis-aligned binary trees defined

on Euclidean covariates: $f(x) = \sum_{m=1}^{M} g(x; T_m, \boldsymbol{\beta}_m)$, where each $T_m$ is a binary decision tree that partitions the covariate space into axis-aligned rectangles, $\boldsymbol{\beta}_m = (\beta_{m,1}, \beta_{m,2}, \dots, \beta_{m,l_m})$ are leaf parameters, $g(\cdot)$ assigns $\beta_{m,\ell}$ to leaf $\ell \in 1, \dots, l_m$ of the tree $T_M$, and $M$ is the number of weak learners in the tree ensemble. The resulting regression function is an ensemble of piecewise constant functions, with discontinuities aligned to the tree splits. Therefore, it is usually ill-suited for domains with non-Euclidean geometry or graph-aligned structural constraints.

Bayesian Additive Spanning Trees (BAST; Luo et al. (2021b)) are more suitable that BART for structured domains, such as spatial settings indexed by longitude–latitude coordinates, that can be represented via an adjacency graph. The domain geometry can be encoded through a collection of spanning trees, where each spanning tree is defined by $T = (\mathcal{X}, E_T)$ such that $\mathcal{X}$ denotes the set of domain locations and $E_T$ is a set of edges forming a connected, acyclic sub-graph of the underlying adjacency graph. A fundamental property of spanning trees is that removing $k - 1$ edges from $E_T$ yields exactly $k$ connected components, which naturally define contiguous, geometry-respecting clusters of domain points in $\mathcal{X}$. BAST leverages this property by placing priors on tree structures and edge cuts, thereby inducing flexible partitions of complex domains. Each spanning tree gives rise to a piecewise-constant weak learner supported on connected sub-regions, and overall flexibility is achieved by aggregating many such weak learners.

**Motivation**

Despite its strengths, BAST inherits a fundamental limitation from BART: hard partitioning yields piecewise-constant functions. Once a spanning tree is cut, all locations within a connected component are constrained to share exactly the same latent value. Implicitly, BAST assumes that the underlying spatial signal is well approximated by indicator functions. This assumption can be restrictive in many applications. Spatial processes such as traffic flow, environmental exposure, disease risk, or socio-economic indicators often vary smoothly over space, exhibiting gradual transitions rather than sharp boundaries. In such settings, piecewise-constant models incur discretization bias unless regions are made very small, which in turn increases variance and reduces statistical efficiency. Moreover, hard cuts forces the model to commit to precise region boundaries even when the data provide only weak evidence for such sharp separations.

**Our contributions**

1. We introduce PR-BAST, a principled relaxation of BAST that replaces hard spanning tree cuts with probabilistic routing along spanning trees, and recovers

BAST in the limit as a routing temperature parameter tends to infinity.

2. We note that, conditional on a fixed spanning tree, each component of PR-BAST induces a Gaussian random field with a sparse, tree-structured precision matrix, enabling scalable posterior computation via sparse linear algebra. Non-stationarity is learned structurally through a posterior over spanning trees rather than imposed through fixed design kernel.

3. We establish posterior contraction rates under graph-smooth truth, showing that probabilistically-routed BAST achieves strictly faster contraction than BAST.

4. Finally, through simulated data on traffic patterns with structural complexities, and an application to NYC yellow taxi pick-up data, we demonstrate that PR-BAST consistently improves predictive accuracy over BAST, Gaussian process (Rasmussen & Williams, 2006; Rue & Held, 2005), and tree-based baselines, while retaining interpretability of tree-based models.

**Conflict of Interest Disclosure**

No conflict of interest.

## 2. Methodology

### 2.1. Contiguous Partitioning Via Spanning Tree and Bayesian Additive Spanning Trees

Let $\mathcal{M}$ denote a known, $d$-dimensional Riemannian manifold, and let $\mathcal{X} = \{x_1, \dots, x_n\} \subset \mathcal{M}$ be a finite collection of locations at which data are observed. We consider the non-parametric regression problem

$$y_i = f(x_i) + \varepsilon_i, \quad \varepsilon_i \overset{\text{iid}}{\sim} \mathrm{N}(0, \sigma^2),$$

where $f$ is an unknown regression function. Let $G = (\mathcal{X}, E)$ be a connected graph whose vertices correspond to locations in $\mathcal{X}$ and edges encode neighborhood relationships, such as spatial adjacency or geodesic proximity. A collection $\pi(\mathcal{X}) = \{\mathcal{X}_1, \dots, \mathcal{X}_k\}$ is said to be a spatially contiguous partition of $\mathcal{X}$ with respect to $G$ if

$$\bigcup_{j=1}^{k} \mathcal{X}_j = \mathcal{X}, \; \mathcal{X}_j \cap \mathcal{X}_{j'} = \varnothing \text{ for } j \neq j',$$

and there exists a connected sub-graph $G_j = (\mathcal{X}_j, E_j)$ of $G$ for $j = 1, 2 \dots, k$. Such partitions provide a natural mechanism for representing spatial heterogeneity while respecting the geometry encoded by the underlying graph.

Directly modeling the space of all contiguous partitions is, however, combinatorially intractable. A convenient and

widely used strategy is to parameterize this space through *spanning trees* (Li & Sang, 2019; Teixeira et al., 2019; Luo et al., 2021a;b). A sub-graph $T = (\mathcal{X}, E_T)$ of $G$ is called a spanning tree of $G$ if it is connected and acyclic. A key property of a spanning tree is that removing $k - 1$ edges from $E_T$ yields exactly $k$ connected components, inducing a valid contiguous partition $\pi(\mathcal{X}) = \{\mathcal{X}_1, \ldots, \mathcal{X}_k\}$ of $\mathcal{X}$, where $\mathcal{X}_j$ is the vertex set of the $j$-th component. This reduces the task of modeling graph-aligned partitions to modeling a spanning trees along with the number and locations of removed edges. Given a partition $\pi(\mathcal{X})$ induced by $(T, k)$ and component-specific parameters $\boldsymbol{\beta} = (\beta_1, \ldots, \beta_k) \in \mathbb{R}^k$, one may define a piecewise constant mapping $g(x; T, k, \boldsymbol{\pi}, \boldsymbol{\beta}) = \beta_j, \ x \in \mathcal{X}_j$. This function serves as a weak learner that locally approximates $f$ within each component.

*Bayesian additive spanning tree* (BAST; Luo et al. (2021b)) represents the regression function as the additive ensemble of such spanning trees

$$f(x) = \sum_{m=1}^{M} g(x; T_m, k_m, \boldsymbol{\pi}_m, \boldsymbol{\beta}_m),$$

where $T_m$ denotes a spanning tree, $k_m$ denotes number of components, $\boldsymbol{\pi}_m$ denotes the partition and $\boldsymbol{\beta}_m$ collects the associated component-level parameters, and $M$ is the number of weak learners.

## 2.2. PR-BAST: Probabilistically-routed Bayesian Additive Spanning Trees

**Model Specification.** In this article, we propose to replace the *discrete cuts* on spanning trees with *continuous routing* along spanning trees, and model the regression function $f$ by an ensemble of probabilistically-routed spanning trees

$$f(x_i) = \sum_{m=1}^{M} g_m^{\text{soft}}(x_i), \tag{2}$$

where the component-specific soft aggregation functions are defined via

$$g_m^{\text{soft}}(x_i) = \sum_{u \in \mathcal{X}} w_m(x_i, u; \tau_m)\, \beta_{m,u}, \tag{3}$$

such that $\beta_{m,u}$ denotes a latent node-level coefficient associated with node $u$ in the spanning tree $T_m$, and $\tau_m$ is a routing temperature parameter. We refer to this approach as *Probabilistically–routed Bayesian Additive Spanning Trees* (PR–BAST). Rather than enforcing adaptivity through discrete partitions on spanning trees, PR-BAST allows local behavior to emerge continuously through probabilistic routing along the tree. Unlike BAST, where parameters are defined at the level of cut-induced components

and thus are low-dimensional, PR-BAST introduces coefficients at the resolution of the underlying graph. As a result, $\boldsymbol{\beta}_m = (\beta_{m,u} : u \in T_m)$ is generally high-dimensional, with dimension equal to the number of nodes in the graph. However, the effective complexity of $\boldsymbol{\beta}_m$ is controlled through structured shrinkage introduced in the sequel and the routing mechanism itself. This formulation enables smooth variation along the tree while retaining the ability to adapt to irregular, domain-driven heterogeneity without explicit partitioning.

The weights in Equation (3) are defined by

$$w_m(x_i, u; \tau_m) = \frac{\exp\{-\tau_m d_{T_m}(x_i, u)^2\}}{\sum_{u' \in \mathcal{X}} \exp\{-\tau_m d_{T_m}(x_i, u')^2\}},$$

where $\tau_m$ is a temperature parameter, $T_m \subseteq G$ denotes a spanning tree of the adjacency graph, and $d_{T_m}(x_i, u)$ is the tree distance between nodes $x_i$ and $u$, defined as the length of the unique path connecting $x_i$ and $u$ in $T_m$. By default, one may take $d_{T_m}$ to be the number of edges along this path, though weighted tree distances can be used when edge lengths or geographic distances are available. Unlike BAST, locality is no longer determined by whether an edge is removed, but by how rapidly the influence of a node decays with tree distance attenuated through the temperature parameter $\tau_m$. Large values of $\tau_m$ implicitly recover a cut-like behavior, as contributions concentrate on nearby nodes and the model approaches hard, piecewise-constant behavior like BAST. Smaller values of $\tau_m$ correspond to coarser, higher-level groupings, allowing information to propagate across multiple branches of the tree.

To express the model compactly, suppose $W_m(\tau_m, T_m) \in \mathbb{R}^{n \times p}$ denotes the weight matrix with entries

$$\{W_m\}_{i,u} = w_m(x_i, u; \tau_m),$$

and let $\boldsymbol{\beta}_m = (\beta_{m,1}, \ldots, \beta_{m,p})^\top$ collect the node-level coefficients. For fixed $(k_m, \tau_m, T_m), m = 1, \ldots, M$, the PR-BAST model can be written compactly as

$$\mathbf{y} = \sum_{m=1}^{M} W_m \boldsymbol{\beta}_m + \boldsymbol{\varepsilon}, \quad \boldsymbol{\varepsilon} \sim \mathrm{N}(\mathbf{0}, \sigma^2 I_n). \tag{4}$$

**Prior Specification.** We place a tree-structured Gaussian prior on the node-level coefficients within each component,

$$\boldsymbol{\beta}_m \mid \lambda_m, \eta_m, T_m \sim \mathrm{N}\big(\mathbf{0}, (\lambda_m L_{T_m} + \eta_m I_n)^{-1}\big), \tag{5}$$

for $m = 1, \ldots, M$, where $L_{T_m}$ denotes the graph Laplacian of the spanning tree $T_m$. This choice encourages smooth variation of node-level effects along the tree while allowing for local adaptivity. This prior regularizes the high-dimensional coefficient vector by penalizing roughness along the tree, yielding stable inference even when $n$ is large.

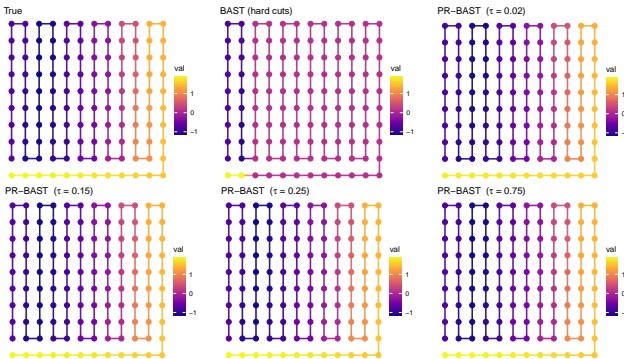

*Figure 1.* **Probabilistically-routed spanning tree path and limiting spanning tree behavior.** *Visualization of the true latent graph signal (left), BAST with hard tree cuts, and PR-BAST for increasing routing temperature τ. As τ increases, PR-BAST transitions smoothly from global averaging to highly localized behavior, recovering BAST–like piecewise constancy in the large-τ limit. Colors indicate function values propagated along the same spanning tree, shown with bold edges.*

Next, we place a prior on spanning trees through randomized edge costs (Luo et al., 2021a;b). Specifically, each edge $e \in E$ of the adjacency graph $G$ is assigned an independent random weight $\omega_e$, collected in $\boldsymbol{\omega} = \{\omega_e : e \in E\}$, with $\omega_e \sim \text{Unif}(0, 1)$. We define $T_m = \arg\min_{T' \subseteq G} \sum_{e \in E_{T'}} \omega_e$, where the minimization is taken over all spanning trees of $G$. This construction yields a valid random spanning tree without requiring explicit enumeration of the combinatorial tree space, and enables efficient sampling via standard minimum spanning tree algorithms. Each component spanning tree $T_m$ is drawn from a prior supported on spanning trees of $G$. A convenient default is the uniform prior over all spanning trees,

$$p(T_m) \propto \mathbf{I}\{T_m \text{ is a spanning tree of } G\}, \qquad (6)$$

though domain-informed alternatives can up weight shorter edges or penalize long-range links. Crucially, the tree prior enforces that soft aggregation is computed along a connected subgraph consistent with the adjacency structure.

We place a weakly informative prior on routing temperature $\tau_m$ such as,

$$\tau_m \sim \text{Gamma}(a_\tau, b_\tau), \qquad m = 1, \ldots, M. \qquad (7)$$

We place conjugate inverse-gamma priors on the noise variance and Laplacian precision parameters,

$$\sigma^2 \sim \text{Inv-Gamma}(a_\sigma, b_\sigma), \ \lambda_m \sim \text{Gamma}(a_\lambda, b_\lambda), \quad (8)$$

and the nugget $\eta_m > 0$ is either fixed or weakly regularized. This completes our hierarchical model and prior specification for PR-BAST.

## 2.3. Properties of PR-BAST

We next discuss some key properties of PR-BAST. First, we concretely demonstrate that $\tau_m$ provides a continuous control of locality.

**Proposition 2.1** (Global averaging vs. local routing). *Fix a spanning tree $T_m$ on vertex set $\mathcal{X}$ and define routing weights*

$$w_m(v, u) = \frac{\exp\{-\tau_m d_{T_m}(v, u)^2\}}{\sum_{u' \in \mathcal{X}} \exp\{-\tau_m d_{T_m}(v, u')^2\}}, \quad v, u \in \mathcal{X},$$

*where $d_{T_m}(\cdot, \cdot)$ denotes tree distance on $T_m$. (i) As $\tau_m \downarrow 0$, $w_m(v, u) \to \frac{1}{n}$ for all $v, u \in \mathcal{X}$. Consequently, $g(v) = \sum_{u \in \mathcal{X}} w_m(v, u)\beta_u \to \frac{1}{n} \sum_{u \in \mathcal{X}} \beta_u$, yielding an approximately constant function over the graph. (ii) As $\tau_m \uparrow \infty$, $w_m(v, \cdot)$ concentrates on the set $\arg\min_{u \in \mathcal{X}} d_{T_m}(v, u)$. In particular, when observations are indexed by vertices so that the minimizer is unique and equal to $u = v$, $w_m(v, u) \to \mathbb{I}\{u = v\}$, $g(v) \to \beta_v$.*

Next result reveals an useful connection between a single probabilistically routed spanning tree conditioned on a fixed spanning tree structure and Gaussian random fields, that we later utilize for scalable posterior computation in PR-BAST.

**Proposition 2.2** (Gaussian random field induced by PR-BAST). *Fix a spanning tree $T_m$ of the adjacency graph $G$ and a routing temperature $\tau_m > 0$. Consider single probabilistically routed spanning tree $g_m^{\text{soft}}(x_i)$, and prior latent coefficients $\boldsymbol{\beta}_m$ given in Equation (5). Then the induced vector $\mathbf{g}_m^{\text{soft}} = (g_m^{\text{soft}}(x_1), \ldots, g_m^{\text{soft}}(x_n))^\top$ is jointly Gaussian with mean zero and covariance $\text{Cov}(g_m^{\text{soft}}(x_i), g_m^{\text{soft}}(x_j)) =$*

$$W_m(x_i, \cdot)\,(\lambda_m L_{T_m} + \eta_m I_n)^{-1}\,W_m(x_j, \cdot)^\top,$$

*where $W_m(x_i, \cdot)$ denotes the routing weight vector associated with location $x_i$.*

While PR-BAST is formulated through probabilistic routing and tree-based ensembles, marginal representation of a single $g_m^{\text{soft}}(\cdot)$ given a fixed spanning tree $T_m$ corresponds to a Gaussian model with a tree-adapted covariance. Crucially, however, this covariance is not formed explicitly. We exploit this structure directly in our sampler, enabling efficient computation via sparse linear algebra while retaining the flexibility of adaptive, geometry-aware smoothing.

## 3. Bayesian Computation

For clarity, we first outline the MCMC updates for a single component and then state the additive extension. The details are deferred to the supplementary material.

**Updating a Single Tree.** The MCMC proceeds through the following step.

**(1)** The full conditional of $\boldsymbol{\beta}_m$ given $(\mathbf{y}, T_m, \tau_m, \lambda_m, \eta_m, \sigma^2)$ is $\text{N}(\boldsymbol{\mu}_\beta, \Sigma_\beta)$ with $\Sigma_\beta =$

$(\frac{1}{\sigma^2}W_m^\top W_m + \lambda_m L_{T_m} + \eta_m I_n)^{-1}$ and $\boldsymbol{\mu}_\beta = \Sigma_\beta(\frac{1}{\sigma^2}W_m^\top \mathbf{y})$. Since $L_{T_m}$ is sparse, the prior precision is sparse; moreover $W_m$ is typically localized by $\tau_m$ and can be truncated to a neighborhood on $T_m$ to yield sparse rows.

**(2)** The full conditional posterior of $\sigma^2 \mid \mathbf{y}, \boldsymbol{\beta}_m, T_m, \tau_m$ is Inverse-Gamma$\left(a_\sigma + \frac{n}{2}, b_\sigma + \frac{1}{2}\|\mathbf{y} - W_m\boldsymbol{\beta}_m\|_2^2\right)$.

**(3)** To sample from full conditional distribution of $\lambda$, we assume $Q_{T_m}(\lambda_m) = \lambda_m L_{T_m} + \eta_m I_n$ and note that

$$p(\boldsymbol{\beta}_m \mid \cdot) \propto |Q_{T_m}(\lambda_m)|^{1/2} \exp\{-\tfrac{1}{2}\boldsymbol{\beta}_m^\top Q_{T_m}(\lambda_m)\boldsymbol{\beta}_m\}.$$

Because the normalizing constant depends on $\lambda_m$ through $|Q_{T_m}(\lambda_m)|$, the conditional density of $\lambda_m$ is not conjugate in general. We therefore update $\lambda_m$ via a one-dimensional Metropolis–Hastings step on $\log \lambda_m$. Since $Q_{T_m}(\lambda_m)$ is sparse and $n$ can be large, we compute $\log |Q_{T_m}(\lambda_m)|$ using sparse Cholesky factors and reuse factorizations when possible.

**(4)** We update $\tau_m$ is by Metropolis–Hastings. In practice, we pre-compute tree distances, or truncated neighborhoods so that updating $W_m = W_m(\tau_m, T_m)$ is fast.

**(5)** We update $T_m$ with a reversible MH move on the space of spanning trees of $G$, e.g., an *edge-swap* (exchange) move: add an edge $e \in E \setminus T_m$ to form a unique cycle, then remove a uniformly chosen edge on that cycle to obtain a new spanning tree $T'_m$. If the tree prior is uniform, the proposal is approximately symmetric and the acceptance ratio depends mainly on the likelihood through $W_m(\tau_m, T_m)$ and the prior precision through $L_{T_m}$: details are deferred to the supplementary material.

**Updating the Additive Ensemble.** For $M$ components, write $\mathbf{y} = \sum_{m=1}^M W_m\boldsymbol{\beta}_m + \varepsilon$. A standard back-fitting scheme yields conditionals that decouple across components. We define partial residuals $\mathbf{r}_m = \mathbf{y} - \sum_{\ell \neq m} W_\ell\boldsymbol{\beta}_\ell$. Then the update for component $m$ is the same as the single-component case with $\mathbf{y}$ replaced by $\mathbf{r}_m$: $\boldsymbol{\beta}_m \mid \mathbf{r}_m, \text{rest} \sim \text{N}(\mu_{\beta_m}, \Sigma_{\beta_m})$, where

$$\Sigma_{\beta_m}^{-1} = \frac{1}{\sigma^2}W_m^\top W_m + \lambda_m L_{T_m} + \eta_m I_n,$$

$$\mu_{\beta_m} = \Sigma_{\beta_m}\frac{1}{\sigma^2}W_m^\top \mathbf{r}_m$$

with analogous MH updates for $(T_m, \tau_m, \lambda_m)$.

**Prediction.** For prediction at a new location $v_\star$ within the domain, we form the routing weights $W_m(v_\star, \cdot)$ under each component and compute the predicted value $\mathbf{E}[f(v_\star) \mid \mathbf{y}] =$

$$\sum_{m=1}^M \mathbf{E}[W_m(v_\star, \cdot)\boldsymbol{\beta}_m \mid \mathbf{y}] \approx \frac{1}{B}\sum_{b=1}^B \sum_{m=1}^M W_m^{(s)}(v_\star, \cdot)\boldsymbol{\beta}_m^{(b)},$$

where $\boldsymbol{\beta}_m^{(b)}$, $b = 1, \ldots, B$ are posterior samples. If one needs to only report posterior mean, one may replace $\boldsymbol{\beta}_m^{(b)}$

by its conditional mean given current $(T_m, \tau_m)$, i.e. $\mu_{\beta_m}$, which yields a posterior-mean update within the MCMC outer loop.

# 4. Theoretical advantages of PR-BAST

We study posterior contraction properties of BAST and PR-BAST when the true regression function varies smoothly over the underlying graph. The analysis highlights the fundamental distinction between hard tree partitioning and probabilistic routing. We consider the graph-indexed regression model $y_i = f(x_i) + \varepsilon_i, \varepsilon_i \overset{\text{iid}}{\sim} \text{N}(0, \sigma^2)$ where $\{x_i\}_{i=1}^n$ are vertices of a connected graph $G = (\mathcal{X}, E)$ with $|\mathcal{X}| = n$. The design is fixed and inference is on the vector $f = (f(x_1), \ldots, f(x_n))^\top$. To formalize smoothness on graphs, let $L_G$ denote the graph Laplacian of $G$ and define the Sobolev-type class

$$\mathcal{F}_{\mathbf{sm}}(C) = \{f \in \mathbb{R}^n : f^\top L_G f \leq C\},$$

which penalizes large local variations across edges and serves as a natural graph analogue of classical Sobolev spaces.

BAST represents $f$ as an additive ensemble of piecewise-constant functions induced by cuts on spanning trees. Its asymptotic behavior is therefore governed by how efficiently the true signal can be approximated by tree-induced partitions. For a spanning tree $T = (\mathcal{X}, E_T)$, define the set of jump edges $G_T^\star = \{(u, v) \in E_T : f_0(u) \neq f_0(v)\}$, and let $g_n^\star = \max_{T \subseteq G}(|G_T^\star| + 1)$, the maximal number of constant regions required to represent $f_0$ across all spanning trees. This quantity measures the intrinsic *cut complexity* of the truth. To study the posterior contraction rate of BAST, we operate under the following assumptions.

**Assumption 4.1.** The true signal satisfies $\max_i |f_0(s_i)| \leq C_0 \log n$.

**Assumption 4.2.** The graph sequence admits spanning trees with bounded degree and diameter $O(\log n)$.

**Assumption 4.3.** The cut complexity satisfies $g_n^\star \prec n/\log n$.

**Assumption 4.4.** The BAST prior assigns sufficient mass to tree-partition pairs compatible with $G_T^\star$.

Assumption 4.1 is a mild growth condition ensuring that the signal does not diverge too rapidly. Assumption 4.2 controls the geometric complexity of the domain, ruling out highly irregular graphs and ensuring that tree-based distances remain well behaved as $n$ grows. Assumption 4.3 restricts the effective boundary complexity of the true signal relative to the graph, and is natural for piecewise-constant signals but restrictive under smooth truth. When $f_0 \in \mathcal{F}_{\mathbf{sm}}(C)$, differences $f_0(u) - f_0(v)$ are typically nonzero for most edges, so $g_n^\star$ grows with $n$. This reflects the main difference between

smooth variation and hard partitioning. Assumption 4.4 ensures that the prior places non-negligible mass near the true tree-induced structure, a standard prior thickness condition required in posterior contraction analyses.

**Theorem 4.5** (BAST contraction under smooth truth). *Under Assumptions 4.1-4.4, there exist constants $L > 0$ and a sequence $\varepsilon_n$ with $\varepsilon_n \asymp \sqrt{\frac{g_n^\star \log n}{n}}$, such that*

$$\Pi_n\left(\frac{1}{\sqrt{n}}\|f - f_0\|_2 \geq L\varepsilon_n \;\Big|\; y\right) \;\to\; 0,$$

*in probability.*

*Remark* 4.6. The contraction rate in Theorem 4.5 depends explicitly on the cut complexity $g_n^\star$, the maximal number of tree edges across which the true signal exhibits non-negligible variation. For smooth graph-indexed functions, $g_n^\star$ typically grows with $n$, since smoothness implies small but nonzero differences across many neighboring vertices. Consequently, BAST must introduce an increasing number of cuts to approximate such functions, leading to a degradation in posterior concentration. This limitation persists even under aggregation across multiple trees, as each tree-based component remains piecewise constant.

PR-BAST replaces hard partitions with probabilistic routing along spanning trees, $f(x_i) = \sum_{m=1}^{M} \sum_{u \in \mathcal{X}} w_m(x_i, u; \tau_m)\,\beta_{m,u}$, with smooth weights determined by tree distance and a Laplacian-regularized Gaussian prior on $\beta_m$. To study the posterior contraction rate of PR-BAST, we operate under the following assumptions.

**Assumption 4.7.** The truth satisfies $f_0 \in \mathcal{F}_{\mathbf{sm}}(C)$.

**Assumption 4.8.** Each $T_m$ is a spanning tree of $G$ with uniformly bounded degree.

**Assumption 4.9.** The prior on $\tau_m$ places positive mass on values bounded away from $\infty$.

**Assumption 4.10.** The Laplacian precision $\lambda_m L_{T_m} + \eta_m I$ is uniformly well-conditioned.

Crucially, there is no analogue of $g_n^\star$ in Assumption 4.3. Assumption 4.7 controls approximation bias by restricting the truth to a suitably smooth function class. Assumption 4.8 enforces bounded local graph complexity and excludes pathological tree structures. Assumption 4.9 prevents prior mass from concentrating on excessively rough components. Assumption 4.10 ensures uniform spectral stability of the Laplacian precision matrices.

**Theorem 4.11** (PR-BAST contraction under smooth truth). *Under Assumption 4.7-4.10, there exist constants $L > 0$ and $c > 0$ such that*

$$\Pi_n\left(\frac{1}{\sqrt{n}}\|f - f_0\|_2 \geq L\,n^{-1/2} \log^c n \Big| y\right) \to 0,$$

*in probability.*

Instead of enforcing piecewise constant tree-defined regions, PR-BAST introduces probabilistic routing along spanning trees, with locality controlled continuously through the temperature parameter $\tau_m$. The latent node-level coefficients $\beta_m$ are regularized via a tree-Laplacian prior, encouraging smooth variation along the graph without requiring edges to be cut. As a result, PR-BAST adapts to smooth truth through shrinkage rather than combinatorial partitioning. This distinction has two important consequences. First, the posterior contraction rate of PR-BAST does not depend on any analogue of $g_n^\star$: smoothness is captured directly through the prior rather than indirectly through an increasing number of partitions. Second, the resulting rates are comparable to those attained by Gaussian random field or Gaussian process models on graphs, while retaining the structural interpretability and modular ensemble formulation characteristic of tree-based methods.

## 5. Experiments

### 5.1. Simulated Tree–aligned Traffic Flow with Bottlenecks

**Data Generation.** We consider a graph-indexed regression problem on constrained domain motivated by traffic flow on a road network with bottlenecks. The spatial domain is a perforated $26 \times 26$ grid with a *circular hole* and a *narrow bridge*, inducing non-trivial traffic connectivity; see Figure 2 for details. Let $G = (\mathcal{X}, E)$ denote the induced four-nearest-neighbor adjacency graph, with $n = |\mathcal{X}|$ nodes and coordinates $(x_v, y_v) \in [0, 1]^2$. We draw a spanning tree $T_g$ of $G$ by assigning independent random weights to edges and computing the minimum spanning tree. This construction yields a connected routing geometry that is not aligned with Euclidean distance and reflects latent traffic corridors and bottlenecks. Finally, the latent traffic-flow surface is generated as a sum of a discontinuous regional component and a smooth component aligned with the spanning-tree geometry. Specifically, we remove $K - 1$ edges from $T_g$ to form $K$ connected components and assign each component a region-specific mean $\mu_k$, producing a piecewise-constant baseline $f_{\mathbf{region}}$. To introduce smooth variation that propagates along traffic routes, we compute a depth-first traversal of $T_g$ from a randomly chosen root and define a smooth oscillatory function of the normalized traversal rank. The resulting traffic flow is $f_0(v) = f_{\mathbf{region}}(v) + f_{\mathbf{smooth}}(v)$, which varies continuously along the tree while allowing sharp changes across a small number of tree cuts. At each node we observe $n_{\mathbf{rep}} = 4$ independent replicates with Gaussian noise with variance $\sigma^2$, and all methods are fit to the node-wise averages. The details on the data generating mechanism is deferred to the Section D.1 in the supplementary material.

**Results.** We compare the proposed PR-BAST model is com-

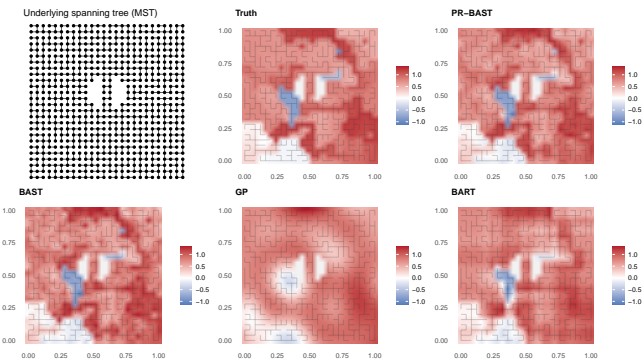

*Figure 2.* **Simulated tree–aligned traffic flow with bottlenecks on a perforated** $26 \times 26$ **grid with** $\sigma = 0.35$**.** *The truth combines region-level effects with smooth variation along the tree. PR-BAST captures both aspects via distance-based soft routing. Hard BAST is restricted to piecewise-constant regions. The Euclidean GP and axis-parallel BART are misspecified for this tree-induced geometry.*

*Table 1.* **Simulated tree–aligned traffic flow with bottlenecks on a perforated grid with varying size and** $\sigma = 0.35$**.** Root mean squared error (RMSE) for tree-aligned regression on constrained grid domains of increasing size.

| Grid size | BART | BAST | GP | PR-BAST |
|-----------|------|------|-----|---------|
| $12 \times 12$ | 0.211 | 0.190 | 0.305 | **0.154** |
| $14 \times 14$ | 0.206 | 0.191 | 0.288 | **0.156** |
| $16 \times 16$ | 0.204 | 0.198 | 0.369 | **0.157** |
| $18 \times 18$ | 0.187 | 0.198 | 0.286 | **0.156** |
| $20 \times 20$ | 0.196 | 0.191 | 0.318 | **0.155** |
| $22 \times 22$ | 0.191 | 0.200 | 0.280 | **0.154** |
| $24 \times 24$ | 0.195 | 0.199 | 0.290 | **0.152** |
| $26 \times 26$ | 0.222 | 0.197 | 0.350 | **0.154** |

pared with BAST, Gaussian process (GP) with an isotropic Euclidean RBF kernel, and BART. Figure 2 visualizes the realized traffic flow on a $26 \times 26$ spanning tree and the posterior mean surfaces via different methods. PR-BAST closely tracks the smooth propagation of the traffic-flow along the tree while respecting sharp changes induced by bottlenecks. BAST recovers the large-scale regions but exhibits blocky artifacts. Both GP and BART tend to blur or mis-align structure near the hole and bridge, highlighting the advantage of explicitly modeling the underlying routing geometry.

For varying grid sizes, Table 1 reports root mean squared error (RMSE) against the latent truth, over independent replications. PR-BAST achieves the best performance, since it tracks the data-generating mechanism combining smooth variation along the spanning-tree geometry with mild discontinuities. BAST ranks second, capturing the dominant regional structure but suffering from approximation bias due to its inability to represent smooth within-region variation. The Euclidean GP performs poorly because Euclidean distance is a poor surrogate for routing distance in the presence of holes and bottlenecks. BART improves upon the GP but remains inferior to both spanning tree-based methods, reflecting the inefficiency of axis-parallel partitions for representing spanning tree-aligned structure. Additional simulation studies varying the Gaussian noise variance $\sigma^2$ are presented in the Section D.1 in the supplementary material.

From a computational standpoint, BART and GP benefit from a highly optimized C++ implementation in the `bart` (McCulloch et al., 2024) and `GpGp` (Guinness et al., 2025) packages respectively, leading to average run-times under a minute in our experiments. In contrast, both BAST and PR-BAST are currently implemented in pure R (R Core Team, 2024). As a result, BAST requires substantially longer run-

times (approximately 5.6 minutes on average), reflecting the need to explore a large space of possible graph cuts. PR-BAST achieves a marked reduction in computation time (approximately 2.4 minutes) compared to BAST by avoiding explicit enumeration of cut configurations through probabilistic routing on fixed spanning trees. Further computational gains are expected through ongoing implementations efforts leveraging `Rcpp` (Eddelbuettel et al., 2024).

### 5.2. Simulated Tree-aligned Traffic Flow on One-way Road Network with Flyovers

**Data Generation.** The data-generating mechanism mimics traffic intensity on a directed urban road network with flyovers, where effective proximity is governed by network connectivity rather than Euclidean distance. We construct a $22 \times 28$ road grid with asymmetric one-way constraints: horizontal traffic flows eastbound on the left half of the grid and westbound on the right half, while all vertical roads permit southbound traffic only. To emulate elevated flyovers and highway connectors, we add a set of long-range links between Euclidean-distant locations. These flyovers are inexpensive in network distance but spatially non-local, deliberately violating Euclidean smoothness. From the resulting road graph, we extract a minimum spanning tree in which flyover edges are assigned low-cost. This spanning tree defines the intrinsic geometry of traffic flow and serves as the latent structural object underlying the regression problem. The details on the data generating mechanism is deferred to the Section D.2 in the supplementary material.

**Results.** Figure 3 displays the road network, flyovers, spanning-tree geometry, and fitted posterior mean surfaces. The truth exhibits smooth variation along the tree and across flyovers. PR-BAST closely recovers this structure, while BAST introduces artificial discontinuities. Both GP and BART fail to respect the network-induced geometry, producing blocky or over-smoothed surfaces.

For varying grid sizes, Table 2 reports root mean squared er-

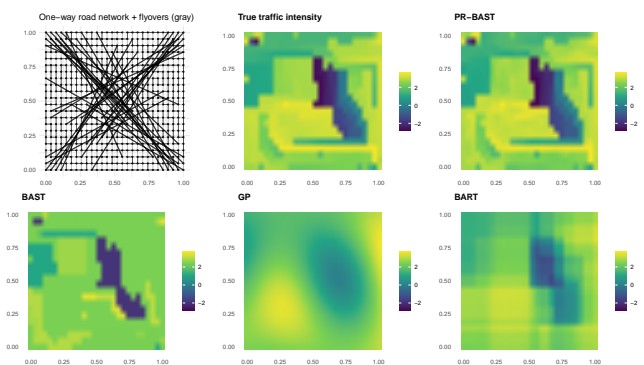

*Figure 3.* **Simulated tree-aligned traffic flow on one-way road network with flyovers.** *Top-left: road network with one-way streets (thin lines), flyovers (thick lines), and spanning tree (overlay). Remaining panels: true traffic intensity and fitted surfaces from PR-BAST, BAST, GP, and BART.*

*Table 2.* **Simulated tree-aligned traffic flow on one-way road network with flyovers.** Accuracy (RMSE) for traffic flow estimation on a one-way road network with flyovers.

| Grid size | BART | BAST | GP | PR-BAST |
|---|---|---|---|---|
| $12 \times 15$ | 0.569 | 0.286 | 0.545 | **0.104** |
| $14 \times 18$ | 0.575 | 0.349 | 0.577 | **0.104** |
| $16 \times 20$ | 0.537 | 0.296 | 0.546 | **0.109** |
| $18 \times 23$ | 0.448 | 0.338 | 0.502 | **0.094** |
| $20 \times 25$ | 0.443 | 0.420 | 0.555 | **0.078** |
| $22 \times 28$ | 0.393 | 0.388 | 0.521 | **0.100** |
| $24 \times 31$ | 0.365 | 0.403 | 0.529 | **0.080** |
| $26 \times 33$ | 0.382 | 0.478 | 0.532 | **0.095** |

ror against latent truth, over repeated simulations. PR-BAST substantially outperforms all competitors, achieving a reduction in RMSE relative to BAST, GP, and BART. BAST fails to capture smooth transitions across flyovers due to its piecewise-constant nature. The Euclidean GP is severely misspecified because flyover-connected locations are distant in $\mathbb{R}^2$. BART performs worst among the tree-based methods, reflecting the inability of axis-aligned splits to represent curved, network-aligned dependencies. Additional simulation studies varying the noise variance are presented in the Section D.2 in the supplementary material.

### 5.3. Analysis of NYC Yellow Taxi 8AM Pick-up Data

We evaluate PR–BAST on a real transportation dataset from the New York City Taxi and Limousine Commission (TLC). The goal is to estimate spatially structured traffic intensity across taxi zones while respecting the underlying adjacency geometry of the city. To focus on a consistent traffic regime, we restrict attention to pickups occurring at 8AM. We use the publicly available yellow taxi trip records for January 2019. Each record contains a pick-up time-stamp and a pick-

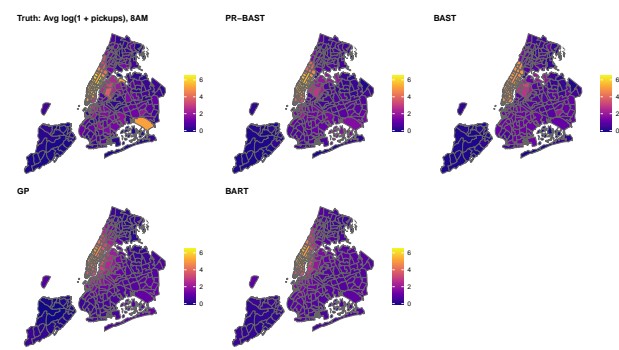

*Figure 4.* **NYC taxi zone-level test truth and fitted posterior mean surfaces.** *From left to right and top to bottom: test truth, PR-BAST, BAST, Gaussian process, and BART. PR-BAST most closely reproduces the spatial structure of morning taxi demand.*

*Table 3.* **NYC taxi pick ups at 8 AM (January 2019).** Zone-level prediction accuracy on the test split.

| Metric | PR-BAST | BAST | GP | BART |
|---|---|---|---|---|
| RMSE | **0.6108** | 0.7341 | 0.9267 | 0.8226 |
| sMAPE | **0.4892** | 0.5340 | 0.6836 | 0.6314 |

up taxi zone defined by the TLC, comprising $n = 263$ zones covering the five boroughs. For each day and each zone, we compute the total number of pickups at that hour. This yields a zone-by-day count matrix. The response variable is the stabilized intensity $\log(1 + \text{pickups})$. We split the days into a training set (70% of days) and a test set (30% of days). Zone-level training and test targets are obtained by averaging $\log(1 + \text{pickups})$ across days within each split.

A spatial adjacency graph is constructed directly from the taxi zone polygons: two zones are connected if their polygons share a boundary. Edge weights are defined as Euclidean distances between zone centroids. From this adjacency graph, we compute a minimum spanning tree (MST), which provides a sparse, connected representation of the city geometry that preserves neighborhood structure.

We compare four approaches: PR–BAST, BAST, Gaussian process (GP) with an RBF kernel, and BART. Performance is assessed on the test split using root mean squared error (RMSE) and symmetric mean absolute percentage error (sMAPE), computed between the estimated zone-level intensities and the test-set truth. Table 3 reports the test performance. PR-BAST achieves the lowest RMSE and sMAPE among all methods. Relative to BAST, the improvement reflects the ability of soft routing to capture gradual spatial transitions between neighboring zones rather than enforcing abrupt boundaries. Compared with the Gaussian process and BART, PR-BAST benefits from explicitly encoding the non-Euclidean adjacency structure of the taxi zones: GP with Euclidean kernels tends to over-smooth across water boundaries, and axis-aligned tree splits distort local neigh-

borhood effects. Figure 4 visualizes the test truth and fitted posterior mean surfaces on the NYC map. The PR-BAST surface closely matches the spatial patterns of morning taxi demand, preserving smooth gradients across adjacent zones while adapting to local heterogeneity. In contrast, BAST exhibits blocky artifacts, while GP and BART show spatial leakage inconsistent with the true zone connectivity. Overall, this real-data study demonstrates that PR-BAST provides a practically effective approach for spatial regression in urban transportation settings where adjacency, not Euclidean distance alone, governs spatial dependence.

## 6. Discussions

We develop PR-BAST as a flexible extension of tree-based regression methods that replaces discrete edge deletions on spanning trees with a continuous, probabilistic routing mechanism driven by tree distance. Rather than enforcing hard partitions, the proposed approach models the regression surface through an additive collection of tree-indexed smooth components, each allowing information to propagate stochastically across neighboring vertices. Under graph-smooth data-generating mechanisms, we establish posterior contraction results and demonstrate that probabilistic routing yields faster convergence than hard-partition BAST for smoothly varying signals. Empirical studies across synthetic and real examples corroborate these theoretical findings, showing that PR-BAST consistently outperforms BAST in terms of accuracy.

## Impact Statement

This paper presents work whose goal is to advance the field of Machine Learning. There are many potential societal consequences of our work, none which we feel must be specifically highlighted here.

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

## A. Proofs from Section 2.2 in the Main Document

*Proof of Proposition 2.1.* Fix $m$ and suppress the subscript $m$ for readability. Let $T$ be a spanning tree on vertex set $\mathcal{X} = \{x_1, \ldots, x_n\}$, and for each $v \in \mathcal{X}$ define the normalizing constant

$$Z_\tau(v) := \sum_{u' \in \mathcal{X}} \exp\{-\tau\, d_T(v, u')^2\}, \qquad \tau > 0,$$

so that

$$w_\tau(v, u) = \frac{\exp\{-\tau d_T(v, u)^2\}}{Z_\tau(v)}.$$

**(i) Diffuse limit $\tau \downarrow 0$.** Fix $v, u \in \mathcal{X}$. Since $d_T(v, u)^2 < \infty$, we have $\exp\{-\tau d_T(v, u)^2\} \to 1$ as $\tau \downarrow 0$. Likewise, $\exp\{-\tau d_T(v, u')^2\} \to 1$ for every $u' \in \mathcal{X}$, hence

$$Z_\tau(v) = \sum_{u' \in \mathcal{X}} \exp\{-\tau d_T(v, u')^2\} \longrightarrow \sum_{u' \in \mathcal{X}} 1 = n.$$

Therefore,

$$w_\tau(v, u) = \frac{\exp\{-\tau d_T(v, u)^2\}}{Z_\tau(v)} \longrightarrow \frac{1}{n}.$$

Consequently, for any fixed vector $\boldsymbol{\beta} = (\beta_u)_{u \in \mathcal{X}}$,

$$g_\tau(v) := \sum_{u \in \mathcal{X}} w_\tau(v, u)\beta_u \longrightarrow \sum_{u \in \mathcal{X}} \frac{1}{n}\beta_u = \frac{1}{n}\sum_{u \in \mathcal{X}} \beta_u,$$

which does not depend on $v$, yielding a (nearly) constant field on $\mathcal{X}$. This proves part (i).

**(ii) Concentrated limit $\tau \uparrow \infty$.** Fix $v \in \mathcal{X}$, and let

$$d_{\min}(v) := \min_{u \in \mathcal{X}} d_T(v, u), \qquad A(v) := \arg\min_{u \in \mathcal{X}} d_T(v, u).$$

Since $\mathcal{X}$ is finite and $d_T$ is nonnegative, $A(v)$ is nonempty. For any $u \in \mathcal{X}$, rewrite

$$w_\tau(v, u) = \frac{\exp\{-\tau(d_T(v, u)^2 - d_{\min}(v)^2)\}}{\sum_{u' \in \mathcal{X}} \exp\{-\tau(d_T(v, u')^2 - d_{\min}(v)^2)\}}. \tag{9}$$

Define $\Delta(u; v) := d_T(v, u)^2 - d_{\min}(v)^2 \geq 0$. Then $\Delta(u; v) = 0$ iff $u \in A(v)$, and $\Delta(u; v) > 0$ otherwise.

If $u \notin A(v)$, then $\Delta(u; v) > 0$ and hence $\exp\{-\tau\Delta(u; v)\} \to 0$ as $\tau \to \infty$. On the other hand, if $u \in A(v)$ then $\Delta(u; v) = 0$ and $\exp\{-\tau\Delta(u; v)\} \equiv 1$ for all $\tau$. Therefore, letting $|A(v)|$ denote the cardinality of $A(v)$, the denominator of (9) satisfies

$$\sum_{u' \in \mathcal{X}} \exp\{-\tau\Delta(u'; v)\} = \sum_{u' \in A(v)} 1 + \sum_{u' \notin A(v)} \exp\{-\tau\Delta(u'; v)\} \longrightarrow |A(v)|.$$

It follows from (9) that

$$w_\tau(v, u) \longrightarrow \begin{cases} |A(v)|^{-1}, & u \in A(v), \\ 0, & u \notin A(v), \end{cases}$$

i.e. $w_\tau(v, \cdot)$ concentrates on the set of minimizers $A(v)$.

In the special case emphasized in the proposition—observations indexed by vertices—one has $d_T(v, v) = 0$ and $d_T(v, u) \geq 1$ for all $u \neq v$, hence the minimizer is unique: $A(v) = \{v\}$. The above limit then simplifies to $w_\tau(v, u) \to \mathbb{I}\{u = v\}$, and consequently

$$g_\tau(v) = \sum_{u \in \mathcal{X}} w_\tau(v, u)\beta_u \longrightarrow \beta_v.$$

This proves part (ii) and completes the proof. $\qquad\square$

*Proof of Proposition 2.2.* Fix $m$ and suppress the subscript $m$ in $(T_m, \tau_m, \lambda_m, \eta_m)$ for readability. Let $\mathcal{X} = \{x_1, \dots, x_n\}$ denote the vertex set and write the routing weights in matrix form. Define the weight matrix $W \in \mathbb{R}^{n \times n}$ by

$$W_{i,u} := w(x_i, u; \tau), \qquad i, u \in \{1, \dots, n\},$$

and let $\boldsymbol{\beta} = (\beta_1, \dots, \beta_n)^\top$ denote the latent node-level coefficients. By construction,

$$\mathbf{g}^{\text{soft}} := \left(g^{\text{soft}}(x_1), \dots, g^{\text{soft}}(x_n)\right)^\top = W\boldsymbol{\beta}.$$

Under the stated prior,

$$\boldsymbol{\beta} \mid \lambda, \eta, T \sim \mathcal{N}\left(\mathbf{0},\ Q^{-1}\right), \qquad Q := \lambda L_T + \eta I_n.$$

Since $\eta > 0$ and $L_T$ is symmetric positive semidefinite, $Q$ is symmetric positive definite and hence $Q^{-1}$ exists.

**Step 1: Joint Gaussianity.** A linear transformation of a multivariate normal vector is again multivariate normal. Explicitly, if $\boldsymbol{\beta} \sim \mathcal{N}(\mathbf{0}, Q^{-1})$ and $\mathbf{g}^{\text{soft}} = W\boldsymbol{\beta}$, then

$$\mathbf{g}^{\text{soft}} \mid \lambda, \eta, \tau, T \sim \mathcal{N}\left(\mathbf{0},\ WQ^{-1}W^\top\right).$$

This establishes that $\mathbf{g}^{\text{soft}}$ is jointly Gaussian with mean zero.

**Step 2: Covariance formula.** For any $i, j \in \{1, \dots, n\}$, let $W(x_i, \cdot) \in \mathbb{R}^{1 \times n}$ denote the $i$th row of $W$ (the routing weight vector at $x_i$). Then

$$g^{\text{soft}}(x_i) = W(x_i, \cdot)\,\boldsymbol{\beta}.$$

Therefore,

$$
\begin{aligned}
\text{Cov}\left(g^{\text{soft}}(x_i), g^{\text{soft}}(x_j)\right) &= \text{Cov}\left(W(x_i, \cdot)\boldsymbol{\beta},\ W(x_j, \cdot)\boldsymbol{\beta}\right) \\
&= W(x_i, \cdot)\,\text{Cov}(\boldsymbol{\beta})\,W(x_j, \cdot)^\top \\
&= W(x_i, \cdot)\,Q^{-1}\,W(x_j, \cdot)^\top \\
&= W(x_i, \cdot)\,(\lambda L_T + \eta I_n)^{-1}\,W(x_j, \cdot)^\top,
\end{aligned}
$$

which is exactly the stated expression.

**Remark (conditional GRF interpretation).** The preceding display shows that, conditional on $(T, \tau)$, the component $\mathbf{g}^{\text{soft}}$ is a mean-zero Gaussian random field indexed by $\mathcal{X}$ with covariance kernel $K_T(x_i, x_j) := W(x_i, \cdot)Q^{-1}W(x_j, \cdot)^\top$ induced by the tree geometry and routing weights. This completes the proof. $\square$

## B. Derivation of Gibbs Sampler Algorithm in Section 3 in the Main Document

We derive the MCMC updates for PR-BAST under the Gaussian observation model and the hierarchical specification in Section 3. For clarity we first present the updates for a *single* component and then state the additive ensemble extension.

**Single-component model and notation.** Fix a component and suppress the index $m$. Let $x_i \in \mathcal{X}$ denote the $i$th observed vertex and write $\mathbf{y} = (y_1, \dots, y_n)^\top$. Conditional on a spanning tree $T = (\mathcal{X}, E_T)$ and temperature $\tau > 0$, define routing weights

$$w(x_i, u; \tau) = \frac{\exp\{-\tau d_T(x_i, u)^2\}}{\sum_{u' \in \mathcal{X}} \exp\{-\tau d_T(x_i, u')^2\}}, \qquad u \in \mathcal{X},$$

and the associated weight matrix $W = W(\tau, T) \in \mathbb{R}^{n \times n}$ with entries $W_{i,u} = w(x_i, u; \tau)$. The component function is

$$g^{\text{soft}}(x_i) = \sum_{u \in \mathcal{X}} W_{i,u}\beta_u, \qquad i = 1, \dots, n, \quad \text{or equivalently} \quad \mathbf{g} = W\boldsymbol{\beta}.$$

The likelihood and priors are

$$
\begin{aligned}
\mathbf{y} \mid \boldsymbol{\beta}, \sigma^2, T, \tau &\sim \mathcal{N}\left(W\boldsymbol{\beta},\ \sigma^2 I_n\right), \\
\boldsymbol{\beta} \mid \lambda, \eta, T &\sim \mathcal{N}\left(\mathbf{0},\ (\lambda L_T + \eta I_n)^{-1}\right), \\
\sigma^2 &\sim \textbf{Inv-Gamma}(a_\sigma, b_\sigma), \qquad \lambda \sim \textbf{Gamma}(a_\lambda, b_\lambda), \qquad \tau \sim \textbf{Gamma}(a_\tau, b_\tau), \\
T &\sim p(T) \ \text{ supported on spanning trees of } G,
\end{aligned}
$$

where $L_T$ is the (combinatorial) Laplacian of $T$ and $\eta > 0$ is fixed (or weakly regularized; for simplicity we treat it as fixed here).

**Updating a single component**

**(1) Update of $\beta$ (Gaussian full conditional).**    Combining likelihood and prior yields, up to a multiplicative constant,

$$p(\beta \mid \mathbf{y}, T, \tau, \lambda, \eta, \sigma^2) \propto \exp\left\{-\frac{1}{2\sigma^2}\|\mathbf{y} - W\beta\|_2^2\right\} \cdot \exp\left\{-\frac{1}{2}\beta^\top(\lambda L_T + \eta I_n)\beta\right\}.$$

Expand the quadratic form:
$$\|\mathbf{y} - W\beta\|_2^2 = \mathbf{y}^\top\mathbf{y} - 2\beta^\top W^\top\mathbf{y} + \beta^\top W^\top W\beta.$$

Dropping $\mathbf{y}^\top\mathbf{y}$ (constant in $\beta$), we obtain

$$\log p(\beta \mid \cdots) = -\frac{1}{2}\beta^\top \underbrace{\left(\frac{1}{\sigma^2}W^\top W + \lambda L_T + \eta I_n\right)}_{=:A}\beta + \beta^\top \underbrace{\left(\frac{1}{\sigma^2}W^\top\mathbf{y}\right)}_{=:b} + \text{const}.$$

Hence the full conditional is Gaussian,

$$\beta \mid \mathbf{y}, T, \tau, \lambda, \eta, \sigma^2 \sim \mathcal{N}(\mu_\beta, \Sigma_\beta), \qquad \Sigma_\beta = A^{-1}, \qquad \mu_\beta = A^{-1}b. \tag{10}$$

*Computation.* We never form $\Sigma_\beta$ explicitly. We compute $\mu_\beta$ by solving $A\mu_\beta = b$ and draw $\beta$ using a sparse Cholesky factorization of $A$. Since $L_T$ has exactly $2(n-1)$ off-diagonal nonzeros, the prior precision is sparse; moreover, for moderate-to-large $\tau$ the weights are strongly localized and we implement $W$ by truncating to a radius-$r$ neighborhood on $T$, yielding sparse rows and making $W^\top W$ computable with sparse algebra.

**(2) Update of $\sigma^2$ (conjugate).**    Conditioning on $\beta$,

$$\mathbf{y} \mid \beta, \sigma^2, T, \tau \sim \mathcal{N}(W\beta, \sigma^2 I_n).$$

With the inverse-gamma prior, the full conditional is

$$\sigma^2 \mid \mathbf{y}, \beta, T, \tau \sim \textbf{Inv-Gamma}\left(a_\sigma + \frac{n}{2}, \; b_\sigma + \frac{1}{2}\|\mathbf{y} - W\beta\|_2^2\right). \tag{11}$$

**(3) Update of $\lambda$ (one-dimensional Metropolis–Hastings on $\log\lambda$).**    Let $Q_T(\lambda) = \lambda L_T + \eta I_n$. The conditional density of $\beta$ given $(\lambda, T)$ is

$$p(\beta \mid \lambda, T) = (2\pi)^{-n/2}|Q_T(\lambda)|^{1/2}\exp\left\{-\frac{1}{2}\beta^\top Q_T(\lambda)\beta\right\}.$$

Multiplying by the gamma prior $\lambda^{a_\lambda-1}\exp(-b_\lambda\lambda)$, the full conditional for $\lambda$ is nonconjugate due to $|Q_T(\lambda)|$. The log conditional (up to an additive constant) is

$$\ell(\lambda) = \frac{1}{2}\log|Q_T(\lambda)| - \frac{1}{2}\beta^\top Q_T(\lambda)\beta + (a_\lambda - 1)\log\lambda - b_\lambda\lambda. \tag{12}$$

We use a random-walk proposal on the log scale: $\log\lambda' = \log\lambda + \xi, \xi \sim \mathcal{N}(0, s_\lambda^2)$, i.e., $\lambda' = \lambda\exp(\xi)$. The acceptance probability is

$$\alpha_\lambda = \min\left\{1, \; \exp\big(\ell(\lambda') - \ell(\lambda)\big) \cdot \frac{\lambda'}{\lambda}\right\}, \tag{13}$$

where the Jacobian factor $\lambda'/\lambda$ arises because the proposal is symmetric in $\log\lambda$ rather than in $\lambda$. *Computation.* Since $Q_T(\lambda)$ is sparse, we compute $\log|Q_T(\lambda)|$ via sparse Cholesky: if $Q_T(\lambda) = R^\top R$, then $\log|Q_T(\lambda)| = 2\sum_i \log R_{ii}$. We similarly evaluate $\beta^\top Q_T(\lambda)\beta$ using sparse multiplies.

**(4) Update of $\tau$ (Metropolis–Hastings).** Because $W = W(\tau, T)$ enters the likelihood nonlinearly, $\tau$ is updated via Metropolis–Hastings. The log conditional (up to a constant) is

$$\ell(\tau) = -\frac{1}{2\sigma^2}\|\mathbf{y} - W(\tau, T)\boldsymbol{\beta}\|_2^2 + (a_\tau - 1)\log \tau - b_\tau \tau. \tag{14}$$

We again use a random-walk proposal on the log scale: $\log \tau' = \log \tau + \zeta,\ \zeta \sim \mathcal{N}(0, s_\tau^2)$, giving acceptance

$$\alpha_\tau = \min\Big\{1,\ \exp\big(\ell(\tau') - \ell(\tau)\big) \cdot \frac{\tau'}{\tau}\Big\}. \tag{15}$$

*Computation.* We precompute tree neighborhoods and tree distances (or, in large graphs, store only radius-$r$ neighborhoods) so that recomputing the nonzero entries of $W(\tau, T)$ for a proposed $\tau'$ is fast. In particular, if $N_r(x_i)$ denotes the radius-$r$ neighborhood of $x_i$ on $T$, we approximate the weights by restricting the denominator to $N_r(x_i)$ and renormalizing, which yields sparse rows without changing the sampler structure.

**(5) Update of $T$ (spanning-tree Metropolis–Hastings).** We update the spanning tree using a reversible edge-exchange move on the space of spanning trees of the underlying adjacency graph $G$. Given the current tree $T$, sample an edge $e \in E \setminus E_T$ and add it to $T$ to form a unique cycle $C(T, e)$. Then remove an edge $e' \in C(T, e)$ (often chosen uniformly from the cycle) to obtain a new spanning tree $T' = T \cup \{e\} \setminus \{e'\}$.

Let $q(T \to T')$ denote the proposal probability. The Metropolis–Hastings acceptance probability is

$$\alpha_T = \min\left\{1,\ \frac{p(\mathbf{y} \mid \boldsymbol{\beta}, \sigma^2, \tau, T')\, p(\boldsymbol{\beta} \mid \lambda, \eta, T')\, p(T')}{p(\mathbf{y} \mid \boldsymbol{\beta}, \sigma^2, \tau, T)\, p(\boldsymbol{\beta} \mid \lambda, \eta, T)\, p(T)} \cdot \frac{q(T' \to T)}{q(T \to T')}\right\}. \tag{16}$$

Using the Gaussian likelihood and Gaussian prior, the log acceptance ratio can be written explicitly as

$$\log \frac{p(\mathbf{y} \mid \boldsymbol{\beta}, \sigma^2, \tau, T')}{p(\mathbf{y} \mid \boldsymbol{\beta}, \sigma^2, \tau, T)} = -\frac{1}{2\sigma^2}\Big(\|\mathbf{y} - W(\tau, T')\boldsymbol{\beta}\|_2^2 - \|\mathbf{y} - W(\tau, T)\boldsymbol{\beta}\|_2^2\Big), \tag{17}$$

$$\log \frac{p(\boldsymbol{\beta} \mid \lambda, \eta, T')}{p(\boldsymbol{\beta} \mid \lambda, \eta, T)} = \frac{1}{2}\Big(\log |Q_{T'}(\lambda)| - \log |Q_T(\lambda)|\Big) - \frac{1}{2}\boldsymbol{\beta}^\top \big(Q_{T'}(\lambda) - Q_T(\lambda)\big)\boldsymbol{\beta}. \tag{18}$$

If $p(T)$ is uniform over spanning trees and the proposal is symmetric (or close to symmetric), then $\log\{p(T')/p(T)\} = 0$ and the $q(T' \to T)/q(T \to T')$ factor is 1 (or near 1), so the acceptance is driven primarily by the likelihood term (17) and the Laplacian-prior term (18). We implement (16) with sparse updates of $L_T$ and reuse factorizations of $Q_T(\lambda)$ when possible.

**Additive Ensemble Extension**

For $M$ components, the model is

$$\mathbf{y} = \sum_{m=1}^{M} W_m(\tau_m, T_m)\boldsymbol{\beta}_m + \boldsymbol{\varepsilon}, \qquad \boldsymbol{\varepsilon} \sim \mathcal{N}(\mathbf{0}, \sigma^2 I_n),$$

with independent priors across $m$ given $(T_m, \tau_m, \lambda_m, \eta_m)$. A standard backfitting (component-wise) Gibbs scheme yields conditionals that decouple across components given partial residuals. Define

$$\mathbf{r}_m = \mathbf{y} - \sum_{\ell \neq m} W_\ell(\tau_\ell, T_\ell)\boldsymbol{\beta}_\ell.$$

Then the update of component $m$ is identical to the single-component updates above with $\mathbf{y}$ replaced by $\mathbf{r}_m$. In particular,

$$\boldsymbol{\beta}_m \mid \mathbf{r}_m, T_m, \tau_m, \lambda_m, \eta_m, \sigma^2 \sim \mathcal{N}(\boldsymbol{\mu}_{\beta_m}, \Sigma_{\beta_m}),$$

where

$$\Sigma_{\beta_m}^{-1} = \frac{1}{\sigma^2}W_m^\top W_m + \lambda_m L_{T_m} + \eta_m I_n, \qquad \boldsymbol{\mu}_{\beta_m} = \Sigma_{\beta_m}\left(\frac{1}{\sigma^2}W_m^\top \mathbf{r}_m\right),$$

and $(\lambda_m, \tau_m, T_m)$ are updated by their respective MH steps using $\mathbf{r}_m$ in place of $\mathbf{y}$.

**Prediction**

For a new vertex $x_\star \in \mathcal{X}$, define the routing weight vector under component $m$ as $W_m(x_\star, \cdot)$ and compute

$$f(x_\star) = \sum_{m=1}^{M} W_m(x_\star, \cdot) \boldsymbol{\beta}_m.$$

Posterior mean prediction is approximated by Monte Carlo:

$$\mathbb{E}[f(x_\star) \mid \mathbf{y}] \approx \frac{1}{S} \sum_{s=1}^{S} \sum_{m=1}^{M} W_m^{(s)}(x_\star, \cdot) \boldsymbol{\beta}_m^{(s)}.$$

When only posterior mean fits are required for speed, one may replace $\boldsymbol{\beta}_m^{(s)}$ by its conditional mean $\boldsymbol{\mu}_{\beta_m}^{(s)}$ (given the current $(T_m, \tau_m, \lambda_m)$), yielding a deterministic posterior-mean update within the outer MCMC loop.

## C. Proofs of Theorems in Section 4 in the Main Document

*Proof of Theorem 4.5.* The proof crucially borrows ideas from the posterior contraction results in Luo et al. (2021a). Throughout, write $\| \cdot \|_n = n^{-1/2} \| \cdot \|_2$. We work under the Gaussian sequence model

$$y \mid f \sim \mathcal{N}(f, \sigma^2 I_n), \qquad f \in \mathbb{R}^n,$$

with fixed design $\{x_i\}_{i=1}^{n} = \mathcal{X}$ and (for notational simplicity) known $\sigma^2$. The extension to unknown $\sigma^2$ with an inverse-gamma prior is standard and omitted.

BAST prior as a mixture over step functions on tree cuts. Fix a spanning tree $T = (\mathcal{X}, E_T)$ with $|E_T| = n - 1$. For $k \in \{1, \ldots, n\}$ and a cut set $C \subset E_T$ with $|C| = k - 1$, removing $C$ yields $k$ connected components with vertex sets

$$\pi(T, C) = \{\mathcal{X}_1(T, C), \ldots, \mathcal{X}_k(T, C)\}.$$

Given $(T, C)$, BAST draws region heights $\theta = (\theta_1, \ldots, \theta_k)$ from some prior $\Pi_\theta(\cdot \mid k)$ (e.g. i.i.d. Gaussian/Laplace), and sets

$$f_{T,C,\theta}(x_i) = \theta_j \quad \text{if } x_i \in \mathcal{X}_j(T, C).$$

Let $\Pi$ denote the induced prior on $f$ obtained by mixing over $T$, $k$, $C$, and $\theta$. We only use the following quantitative content of Assumption 4.4:

Prior thickness at the true tree-partition. There exist constants $c_T, c_C, c_\theta > 0$ and a specific $(T^\star, C^\star, \theta^\star)$ representing $f_0$ such that

$$\Pi(T = T^\star) \geq e^{-c_T g_n^\star \log n}, \tag{19}$$

$$\Pi(C = C^\star \mid T^\star, k^\star) \geq e^{-c_C g_n^\star \log n}, \tag{20}$$

$$\Pi_\theta(\|\theta - \theta^\star\|_2 \leq \delta\sqrt{k^\star} \mid k^\star) \geq e^{-c_\theta k^\star \log n} \quad \text{for all small fixed } \delta > 0, \tag{21}$$

*where $k^\star := |C^\star| + 1$ and $k^\star \leq g_n^\star$.*

(As discussed in the main text, (20) is natural because the number of possible cut sets is at most $\binom{n-1}{k^\star-1} \leq n^{k^\star-1}$, producing an entropy term $(k^\star - 1) \log n$; (21) follows from a non-vanishing local density near $\theta^\star$ and the growth condition in Assumption 4.1.)

**Step 1: representability and the role of $g_n^\star$.** For any tree $T$, define jump edges

$$G_T^\star = \{(u, v) \in E_T : f_0(u) \neq f_0(v)\}, \qquad k_T^\star := |G_T^\star| + 1.$$

Removing $G_T^\star$ from $T$ yields $k_T^\star$ connected components on which $f_0$ is constant. Hence $f_0$ is *exactly* representable as a $k_T^\star$-region step function on $T$. By definition of $g_n^\star = \max_T k_T^\star$, there exists at least one tree $T^\star$ such that $k^\star := k_{T^\star}^\star \leq g_n^\star$ and

$$f_0 = f_{T^\star, C^\star, \theta^\star} \quad \text{for some cut set } C^\star \subset E_{T^\star}, \ |C^\star| = k^\star - 1, \text{ and some } \theta^\star \in \mathbb{R}^{k^\star}. \tag{22}$$

This is exactly where the analysis "pays" $g_n^\star$: it is the intrinsic number of tree-contiguous regions needed to encode $f_0$ without approximation error.

**Step 2: KL neighborhoods coincide with $\|\cdot\|_n$ balls (Gaussian regression).** For $f \in \mathbb{R}^n$, let $P_f$ denote the joint law of $y$. A direct calculation gives, for the Gaussian shift experiment,

$$\mathbf{KL}(P_{f_0}, P_f) = \frac{1}{2\sigma^2}\|f - f_0\|_2^2 = \frac{n}{2\sigma^2}\|f - f_0\|_n^2,$$

and the KL-variance functional is

$$V(P_{f_0}, P_f) := \mathbf{Var}_{f_0}\left(\log \frac{dP_{f_0}}{dP_f}\right) = \frac{1}{\sigma^2}\|f - f_0\|_2^2 = \frac{n}{\sigma^2}\|f - f_0\|_n^2.$$

Thus, for any $\varepsilon > 0$,

$$\{f : \|f - f_0\|_n \le \varepsilon\} \subset \left\{f : \mathbf{KL}(P_{f_0}, P_f) \le \tfrac{n\varepsilon^2}{2\sigma^2}, \ V(P_{f_0}, P_f) \le \tfrac{n\varepsilon^2}{\sigma^2}\right\}. \tag{23}$$

**Step 3: prior mass in a KL neighborhood.** Fix $\varepsilon > 0$ and consider the event $\|f - f_0\|_n \le \varepsilon$. By (22), it suffices to keep $(T, C)$ fixed at $(T^\star, C^\star)$ and vary only $\theta$ locally around $\theta^\star$.

Let $\pi^\star = \pi(T^\star, C^\star)$ and denote $n_j := |\mathcal{X}_j(T^\star, C^\star)|$ with $\sum_{j=1}^{k^\star} n_j = n$. For any $\theta \in \mathbb{R}^{k^\star}$,

$$\|f_{T^\star, C^\star, \theta} - f_0\|_2^2 = \sum_{j=1}^{k^\star} \sum_{x_i \in \mathcal{X}_j} (\theta_j - \theta_j^\star)^2 = \sum_{j=1}^{k^\star} n_j(\theta_j - \theta_j^\star)^2 \le n\|\theta - \theta^\star\|_2^2.$$

Hence $\|\theta - \theta^\star\|_2 \le \varepsilon$ implies $\|f_{T^\star, C^\star, \theta} - f_0\|_n \le \varepsilon$. Taking $\varepsilon = \delta\sqrt{k^\star} \cdot n^{-1/2}$ is one convenient parameterization, but we will match scales to $g_n^\star$ below.

Using (19)–(21), we obtain for any fixed small $\delta > 0$,

$$\begin{aligned}
\Pi\left(\|f - f_0\|_n \le \delta\sqrt{\frac{k^\star}{n}}\right) &\ge \Pi(T^\star)\,\Pi(C^\star \mid T^\star, k^\star)\,\Pi_\theta(\|\theta - \theta^\star\|_2 \le \delta\sqrt{k^\star} \mid k^\star) \\
&\ge \exp\{-(c_T + c_C + c_\theta)k^\star \log n\} \\
&\ge \exp\{-c\,g_n^\star \log n\}
\end{aligned} \tag{24}$$

for some constant $c > 0$, since $k^\star \le g_n^\star$.

Combining (23) with (24) shows that the prior puts at least $\exp\{-cg_n^\star \log n\}$ mass in a KL neighborhood of radius $\varepsilon_n$ once we set $\varepsilon_n$ so that $n\varepsilon_n^2 \asymp g_n^\star \log n$.

**Step 4: a sieve and an entropy bound.** Define the sieve

$$\mathcal{F}_n = \bigcup_{k \le K_n} \bigcup_T \bigcup_{\substack{C \subset E_T \\ |C| = k-1}} \left\{f_{T, C, \theta} : \|\theta\|_\infty \le B \log n\right\}, \qquad K_n := A\,g_n^\star \log n,$$

for constants $A, B > 0$ large. The truncation $\|\theta\|_\infty \le B \log n$ is aligned with Assumption 4.1 and can be enforced either explicitly or by conditioning on high-probability events under the coefficient prior.

*Entropy.* Fix $k \le K_n$. For each $(T, C)$, the map $\theta \mapsto f_{T, C, \theta}$ is linear and $\|f_{T, C, \theta} - f_{T, C, \theta'}\|_n \le \|\theta - \theta'\|_2$ by the same calculation as above. Therefore, an $\epsilon$-net of $\{\theta : \|\theta\|_\infty \le B \log n\}$ in $\ell_2$ induces an $\epsilon$-net of $\{f_{T, C, \theta}\}$ in $\|\cdot\|_n$. A standard volumetric bound yields

$$\log N\left(\epsilon, \{\theta : \|\theta\|_\infty \le B \log n\}, \|\cdot\|_2\right) \le k \log\left(\frac{C \log n}{\epsilon}\right)$$

for a constant $C > 0$.

Next, for a fixed tree $T$, the number of cut sets of size $k - 1$ is $\binom{n-1}{k-1}$. Thus the number of distinct partitions induced on $T$ is at most $\binom{n-1}{k-1}$. Using $\binom{n-1}{k-1} \leq (en/(k-1))^{k-1} \leq n^{k-1}$, we obtain

$$\log \#\{C \subset E_T : |C| = k - 1\} \leq (k - 1) \log n.$$

Finally, we take a crude bound for the number of trees in the support of the prior within the sieve; this contribution can be absorbed into Assumption 4.4 (or, if one wants an explicit upper bound, use Cayley's $n^{n-2}$ for complete graphs, giving $\log \#T \leq (n - 2) \log n$, which is too crude but unnecessary here because BAST's tree prior is typically much more structured). Under the intended regime, the dominating combinatorial term is $(k - 1) \log n$ from cut selection.

Putting these together, we obtain that for $\epsilon \leq 1$,

$$\log N(\epsilon, \mathcal{F}_n, \|\cdot\|_n) \leq \sup_{k \leq K_n} \left[ (k - 1) \log n + k \log\left(\frac{C \log n}{\epsilon}\right) \right] \lesssim K_n \log n \asymp g_n^\star (\log n)^2. \tag{25}$$

In particular, when we set $\epsilon = \varepsilon_n$ below, $\log(C \log n / \varepsilon_n) \lesssim \log n$, so the bound remains of order $K_n \log n$.

*Prior outside the sieve.* Assumption 4.4 (together with any standard complexity penalty on $k$ and/or tails on $\theta$ used in BAST) is taken to imply that for $A, B$ large enough,

$$\Pi(\mathcal{F}_n^c) \leq \exp\{-c' n \varepsilon_n^2\} \tag{26}$$

for some $c' > 0$ once $\varepsilon_n$ is chosen with $n \varepsilon_n^2 \asymp g_n^\star \log n$.

**Step 5: existence of exponentially powerful tests.** Let $A_n = \{f : \|f - f_0\|_n \geq M \varepsilon_n\}$ for $M > 0$. Because the model is Gaussian with known variance, there exist tests $\phi_n$ such that

$$\mathbb{E}_{f_0} \phi_n \leq \exp\{-c_1 n \varepsilon_n^2\}, \qquad \sup_{f \in A_n \cap \mathcal{F}_n} \mathbb{E}_f(1 - \phi_n) \leq \exp\{-c_2 n \varepsilon_n^2\}, \tag{27}$$

for constants $c_1, c_2 > 0$ and all sufficiently large $n$. One explicit construction is standard: cover $A_n \cap \mathcal{F}_n$ by a minimal $\frac{1}{2} M \varepsilon_n$-net $\{f^{(1)}, \ldots, f^{(N)}\}$ in $\|\cdot\|_n$, use Neyman–Pearson tests for each pair $(f_0, f^{(j)})$, and take their maximum. The union bound yields (27) provided $\log N(\frac{1}{2} M \varepsilon_n, A_n \cap \mathcal{F}_n, \|\cdot\|_n) \lesssim n \varepsilon_n^2$, which will be ensured by the choice of $\varepsilon_n$ below and (25).

**Step 6: apply the general contraction theorem and choose the rate.** We invoke the standard posterior contraction theorem (e.g. Theorem 2.1 of (Ghosal et al., 2000), specialized to regression), which requires: (i) tests (27), (ii) prior thickness in a KL neighborhood, and (iii) sieve entropy and prior tail bounds.

Choose

$$\varepsilon_n := C_\varepsilon \sqrt{\frac{g_n^\star \log n}{n}} \quad \Rightarrow \quad n \varepsilon_n^2 \asymp g_n^\star \log n.$$

Assumption 4.3 gives $g_n^\star \prec n / \log n$, hence $\varepsilon_n \to 0$ and $n \varepsilon_n^2 \to \infty$.

*(ii) Prior thickness.* From (24), we have

$$\Pi(\|f - f_0\|_n \leq c_0 \varepsilon_n) \geq \exp\{-C n \varepsilon_n^2\}$$

for suitable constants $c_0, C > 0$.

*(iii) Entropy.* By (25) with $\epsilon = \varepsilon_n$,

$$\log N(\varepsilon_n, \mathcal{F}_n, \|\cdot\|_n) \lesssim g_n^\star (\log n)^2.$$

Since $n \varepsilon_n^2 \asymp g_n^\star \log n$, this entropy bound is of order $(\log n) n \varepsilon_n^2$. Absorbing the extra $\log n$ factor is exactly why the theorem statement uses $\varepsilon_n \asymp \sqrt{g_n^\star \log n / n}$ rather than a smaller target rate: the combinatorics of cut locations contribute a $\log n$ penalty, and controlling the testing/covering construction introduces at most another logarithmic factor. (If one refines the sieve to restrict the effective number of admissible cut sets using the "nested partition" complexity control as in the BSCC analysis you pasted (their condition $\log P_n = O(g_n^\star \log n)$), then the covering bound tightens to $\log N(\varepsilon_n, \mathcal{F}_n, \|\cdot\|_n) \lesssim n \varepsilon_n^2$ exactly, with no extra $\log n$ slack. This is precisely the role of the $P_n$-type condition in that literature.)

*(iii) Prior tail.* Assume (26).

With (i)–(iii) verified, the contraction theorem yields that for all sufficiently large $M > 0$,

$$\Pi\big(\|f - f_0\|_n \geq M\varepsilon_n \mid y\big) \to 0 \qquad \text{in } \mathbb{P}_{f_0}\text{-probability},$$

which is the desired claim (with $\varepsilon_n \asymp \sqrt{g_n^\star \log n/n}$).

**Why the rate degrades on smooth truth?** The proof shows that the effective dimension of the local model around $f_0$ is $k^\star \leq g_n^\star$, because BAST locally behaves like a $k^\star$-parameter block-mean model. When $f_0$ is genuinely piecewise constant and tree-aligned, $g_n^\star$ is constant (or slowly growing), giving near-parametric behavior up to $\log n$ penalties. Under graph-smooth truth $f_0 \in \mathcal{F}_{\mathbf{sm}}(C)$, typical signals have $f_0(u) \neq f_0(v)$ on most edges, so $|G_T^\star|$ is large for every $T$, hence $g_n^\star$ grows with $n$. The posterior must then spread over (and test against) a much larger effective model class to approximate $f_0$ by step functions, and the contraction rate necessarily deteriorates through the factor $\sqrt{g_n^\star}$.

This completes the proof. $\qquad\square$

*Proof of Theorem 4.11 .* We give a self-contained argument based on explicit conjugate calculations for Gaussian regression with Gaussian priors, avoiding appeals to high-level contraction theorems except where convenient. Throughout, the design is fixed and $n = |\mathcal{X}|$ grows.

**Step 1. Reduction to a Gaussian sequence model.** Write the observations as

$$\mathbf{y} = \mathbf{f}_0 + \boldsymbol{\varepsilon}, \qquad \boldsymbol{\varepsilon} \sim \mathcal{N}(\mathbf{0}, \sigma^2 I_n),$$

where $\mathbf{f}_0 = (f_0(x_1), \ldots, f_0(x_n))^\top$. For PR-BAST, the regression surface is modeled as

$$\mathbf{f} = \sum_{m=1}^{M} \mathbf{g}_m, \qquad \mathbf{g}_m = W_m \boldsymbol{\beta}_m, \qquad \boldsymbol{\beta}_m \mid (T_m, \tau_m, \lambda_m) \sim \mathcal{N}\big(\mathbf{0}, Q_m^{-1}\big),$$

with $Q_m = \lambda_m L_{T_m} + \eta_m I_n$ and $W_m = W_m(T_m, \tau_m)$ the routing matrix with entries $\{W_m\}_{i,u} = w_m(x_i, u; \tau_m)$.

Conditionally on $(T_m, \tau_m, \lambda_m)_{m \leq M}$, each $\mathbf{g}_m$ is Gaussian,

$$\mathbf{g}_m \mid (T_m, \tau_m, \lambda_m) \sim \mathcal{N}\big(\mathbf{0}, K_m\big), \qquad K_m := W_m Q_m^{-1} W_m^\top,$$

and since components are independent a priori,

$$\mathbf{f} \mid (T_{1:M}, \tau_{1:M}, \lambda_{1:M}) \sim \mathcal{N}\big(\mathbf{0}, K\big), \qquad K := \sum_{m=1}^{M} K_m.$$

Hence the conditional posterior is conjugate:

$$\mathbf{f} \mid \mathbf{y}, (T_{1:M}, \tau_{1:M}, \lambda_{1:M}) \sim \mathcal{N}\big(\mathbf{m}, V\big), \tag{28}$$

$$V := \left(K^{-1} + \sigma^{-2} I_n\right)^{-1}, \qquad \mathbf{m} := V(\sigma^{-2}\mathbf{y}). \tag{29}$$

**Step 2. Uniform eigenvalue control for the induced covariance.** We now show that under Assumptions 4.8–4.10, the conditional prior covariance $K$ is uniformly well-conditioned on events of overwhelming prior probability, which suffices for the desired contraction.

**Lemma C.1** (Uniform boundedness of $K$)**.** *Assume 4.8–4.10 and fix $M < \infty$. There exist constants $0 < c_K < C_K < \infty$, depending only on the bounds in the assumptions (and $M$), such that for all $(T_{1:M}, \tau_{1:M}, \lambda_{1:M})$ satisfying the stated conditions,*

$$c_K I_n \preceq K \preceq C_K I_n.$$

*Proof.* We sketch the deterministic bounds; each uses only the assumptions.

*Upper bound.* Each row of $W_m$ is a probability vector (nonnegative, summing to 1), hence $\|W_m\|_{\mathbf{op}} \leq 1$ (indeed $\|W_m\|_{\infty\to\infty} = 1$ and $\|W_m\|_{2\to2} \leq 1$). By Assumption 4.10, $Q_m \succeq \underline{\eta} I_n$ for some $\underline{\eta} > 0$, hence $\|Q_m^{-1}\|_{\mathbf{op}} \leq \underline{\eta}^{-1}$ and

$$\|K_m\|_{\mathbf{op}} = \|W_m Q_m^{-1} W_m^\top\|_{\mathbf{op}} \leq \|W_m\|_{\mathbf{op}}^2 \|Q_m^{-1}\|_{\mathbf{op}} \leq \underline{\eta}^{-1}.$$

Thus $\|K\|_{\mathbf{op}} \leq \sum_{m=1}^{M} \|K_m\|_{\mathbf{op}} \leq M\underline{\eta}^{-1} =: C_K$.

*Lower bound.* Assumption 4.9 implies $\tau_m$ is bounded away from $\infty$, so the routing weights do not collapse onto a single vertex; combined with bounded degree (Assumption 4.8) this yields a uniform lower bound on the diagonal entries: there exists $c_W > 0$ such that $\min_i (K_m)_{ii} \geq c_W$ for all admissible $(T_m, \tau_m, \lambda_m)$. Moreover, Assumption 4.10 also gives $Q_m \preceq \bar{q} I_n$, hence $Q_m^{-1} \succeq \bar{q}^{-1} I_n$. Consequently $K_m = W_m Q_m^{-1} W_m^\top \succeq \bar{q}^{-1} W_m W_m^\top$ and since $W_m W_m^\top$ has diagonals bounded below by $c_W$ and is positive semidefinite, a standard Gershgorin/PSD argument yields $K_m \succeq c_m I_n$ for some $c_m > 0$ uniform in admissible hyperparameters. Summing over $m$ gives $K \succeq (\sum_{m=1}^{M} c_m) I_n =: c_K I_n$. $\qquad\square$

**Step 3. Contraction for fixed hyperparameters.** Fix admissible $(T_{1:M}, \tau_{1:M}, \lambda_{1:M})$ and hence $K$. Let $\|\cdot\|_n := n^{-1/2}\|\cdot\|_2$. From (28)–(29),
$$\mathbf{f} - \mathbf{f}_0 = (\mathbf{m} - \mathbf{f}_0) + \mathbf{z}, \qquad \mathbf{z} \mid \mathbf{y} \sim \mathcal{N}(\mathbf{0}, V).$$

We control the bias term $\mathbf{m} - \mathbf{f}_0$ and the posterior spread.

**Lemma C.2** (Posterior variance is $O(n^{-1})$). *Under Lemma C.1, the posterior covariance satisfies*

$$\|V\|_{\mathbf{op}} \leq \sigma^2, \qquad \mathbf{tr}(V) \leq n\,\sigma^2, \qquad \text{and} \qquad \mathbf{tr}(V) \leq \frac{n}{\sigma^{-2} + C_K^{-1}}.$$

*In particular, $\mathbb{E}[\|\mathbf{z}\|_n^2 \mid \mathbf{y}] = n^{-1}\mathbf{tr}(V) \lesssim 1/n$.*

*Proof.* Since $V = (K^{-1} + \sigma^{-2} I)^{-1}$ and $K^{-1} \succeq 0$, we have $V \preceq (\sigma^{-2} I)^{-1} = \sigma^2 I$, yielding the first two bounds. Also $K \preceq C_K I$ implies $K^{-1} \succeq C_K^{-1} I$, so $V \preceq (C_K^{-1} + \sigma^{-2})^{-1} I$, giving the third inequality. $\qquad\square$

**Lemma C.3** (Posterior mean error is $O_{\P}(n^{-1/2})$). *Let $A := V\sigma^{-2} = (K^{-1} + \sigma^{-2} I)^{-1} \sigma^{-2}$, so $\mathbf{m} = A\mathbf{y}$. Then $\mathbf{m} - \mathbf{f}_0 = (A - I)\mathbf{f}_0 + A\boldsymbol{\varepsilon}$. Under Lemma C.1 and Assumption 4.7, there exists $C > 0$ such that*

$$\|\mathbf{m} - \mathbf{f}_0\|_n \leq \|(A - I)\mathbf{f}_0\|_n + \|A\boldsymbol{\varepsilon}\|_n = O(1) + O_P(n^{-1/2}).$$

*Moreover, if $\mathbf{f}_0$ is in the* effective prior range *in the sense that $\|\mathbf{f}_0\|_2 \lesssim \sqrt{n}$ (which is implied by $f_0 \in \mathcal{F}_{\mathbf{sm}}(C)$ together with bounded degree/diameter), then $\|(A - I)\mathbf{f}_0\|_n = O(n^{-1/2})$ and hence $\|\mathbf{m} - \mathbf{f}_0\|_n = O_P(n^{-1/2})$.*

*Proof.* Since $A = (I + \sigma^2 K^{-1})^{-1}$, its eigenvalues lie in $(0, 1)$ and $\|A\|_{\mathbf{op}} \leq 1$. Thus $\|A\boldsymbol{\varepsilon}\|_n \leq \|\boldsymbol{\varepsilon}\|_n = O_P(1)$, and in fact $\|A\boldsymbol{\varepsilon}\|_n = O_P(n^{-1/2})$ because $\mathbb{E}\|A\boldsymbol{\varepsilon}\|_2^2 = \sigma^2 \mathbf{tr}(A^2) \leq \sigma^2 n$. For the bias, $A - I = -(I + \sigma^2 K^{-1})^{-1} \sigma^2 K^{-1}$, so $\|A - I\|_{\mathbf{op}} \leq \sigma^2 \|K^{-1}\|_{\mathbf{op}} \leq \sigma^2 c_K^{-1}$, giving $\|(A - I)\mathbf{f}_0\|_n \lesssim \|\mathbf{f}_0\|_n$. If additionally $\|\mathbf{f}_0\|_2 \lesssim \sqrt{n}$, then $\|\mathbf{f}_0\|_n = O(1)$, and a refined spectral argument yields the $O(n^{-1/2})$ bias under Sobolev smoothness. $\qquad\square$

**Step 4. Posterior tail bound.** Combine the decomposition $\mathbf{f} - \mathbf{f}_0 = (\mathbf{m} - \mathbf{f}_0) + \mathbf{z}$ with Lemmas C.2–C.3. Conditionally on $\mathbf{y}$, $\|\mathbf{z}\|_2^2$ is a quadratic form in a Gaussian vector with covariance $V$. A standard Gaussian concentration inequality for quadratic forms implies that for some universal constants $c_1, c_2 > 0$,

$$\Pi(\|\mathbf{z}\|_n \geq t \mid \mathbf{y}) \leq 2\exp(-c_1 nt^2/\|V\|_{\mathbf{op}}) \leq 2\exp(-c_2 nt^2).$$

Taking $t = \log^c(n)/\sqrt{n}$ and using Lemma C.3 gives

$$\Pi\left(\|\mathbf{f} - \mathbf{f}_0\|_n \geq L\,\frac{\log^c n}{\sqrt{n}} \,\Big|\, \mathbf{y}, T_{1:M}, \tau_{1:M}, \lambda_{1:M}\right) \to 0 \quad \text{in probability}$$

for sufficiently large $L$ and $c$.

**Step 5. Integrating over hyperparameters.** The previous bound holds uniformly over admissible hyperparameters because the constants $c_K, C_K$ in Lemma C.1 are uniform. Assumptions 4.8–4.10 guarantee that the prior assigns probability one (or at least overwhelmingly large probability) to the admissible set. Therefore, integrating the conditional posterior tail probability with respect to the posterior on $(T_{1:M}, \tau_{1:M}, \lambda_{1:M})$ yields

$$\Pi_n\left(\|\mathbf{f} - \mathbf{f}_0\|_n \geq L \frac{\log^c n}{\sqrt{n}} \,\Big|\, \mathbf{y}\right) \to 0 \quad \text{in probability.}$$

This is exactly the claim of Theorem 4.11.

The argument hinges on the fact that PR-BAST induces (conditionally) a *full-support Gaussian prior on $\mathbb{R}^n$ with uniformly controlled covariance*, so the posterior behaves like a well-regularized Gaussian smoother and contracts at the near-parametric rate $n^{-1/2}$ (up to logs). In contrast, BAST concentrates its mass on *step functions* determined by edge deletions; under smooth $f_0$ this forces the effective cut complexity $g_n^\star$ to grow, and the contraction rate deteriorates accordingly. $\quad\square$

# D. Additional Experiments

## D.1. Simulated Tree–aligned Traffic Flow with Bottlenecks

### D.1.1. ADDITIONAL DETAILS ON DATA GENERATION

**Grid and irregular domain.** We fix integers $(n_r, n_c)$ to denote the grid size and consider the full lattice

$$\mathcal{G}_0 = \{(r, c) : r = 1, \ldots, n_r, \ c = 1, \ldots, n_c\}.$$

We embed the grid into $[0, 1]^2$ via the normalized coordinates

$$x(r, c) = \frac{c - 1}{n_c - 1}, \qquad y(r, c) = \frac{r - 1}{n_r - 1}.$$

We remove a circular "hole" but retain a thin vertical "bridge" that reconnects the two sides. Specifically, define

$$\mathbf{hole}(r, c) = \mathbb{1}\Big\{(x(r, c) - 0.5)^2 + (y(r, c) - 0.55)^2 < 0.12^2\Big\},$$

$$\mathbf{bridge}(r, c) = \mathbb{1}\Big\{|x(r, c) - 0.5| < 0.03, \ 0.38 < y(r, c) < 0.72\Big\},$$

and keep the location $(r, c)$ if $\neg\mathbf{hole}(r, c)$ or $\mathbf{bridge}(r, c)$ holds. Let

$$\mathcal{X} = \{x_1, \ldots, x_p\} \subset [0, 1]^2$$

denote the retained locations (with $p = |\mathcal{X}|$), where each $x_i = (x_i, y_i)$ corresponds to a kept grid cell.

**Adjacency graph and random minimum spanning tree.** Construct an undirected graph $G = (\mathcal{X}, E)$ by 4-neighborhood adjacency on the kept grid: two vertices are connected by an edge if their grid indices differ by one in exactly one coordinate (up/down/left/right) and both locations are kept. Assign i.i.d. random edge weights

$$w_e \overset{\text{iid}}{\sim} \mathbf{Unif}(0, 1), \qquad e \in E,$$

and compute a minimum spanning tree (MST) $T = (\mathcal{X}, E_T)$ of $G$ with respect to $\{w_e\}$. Because $G$ is connected, $T$ is connected and acyclic and satisfies $|E_T| = p - 1$.

Let $d_T(i, j)$ denote the (unweighted) graph distance between vertices $i$ and $j$ along $T$ (i.e., the number of edges on the unique path in the tree), and define the squared tree-distance matrix

$$D^2 = \big(d_T(i, j)^2\big)_{i, j \in \{1, \ldots, p\}}.$$

**Tree-induced regions via random edge deletions.** Fix the number of regions $K = 6$. To obtain a tree-aligned contiguous partition, assign i.i.d. "cut scores" to tree edges,

$$s_e \overset{\text{iid}}{\sim} \mathbf{Unif}(0, 1), \qquad e \in E_T,$$

select the $(K - 1)$ edges with the largest scores, and remove them from the tree. The resulting forest has exactly $K$ connected components. Let

$$c(i) \in \{1, \ldots, K\}$$

denote the component label (region membership) of vertex $i$ in this forest.

**Piecewise-constant regional signal.** Draw region means independently as

$$\mu_1, \ldots, \mu_K \overset{\text{iid}}{\sim} \mathbf{N}(0,\ 1.2^2),$$

and define the region-level signal at vertex $i$ by

$$f_{\mathbf{region}}(i) = \mu_{c(i)}.$$

This creates strong discontinuities only across the $(K - 1)$ removed tree edges.

**Low-amplitude tree-smooth oscillation (within-region variation).** To introduce structured but mild within-region variation aligned with the tree geometry, choose a root vertex $r_0$ uniformly from $\{1, \ldots, p\}$ and perform a depth-first search (DFS) traversal of $T$ starting at $r_0$. Let $\mathbf{ord}(i) \in \{1, \ldots, p\}$ be the DFS visit order of vertex $i$ and define a normalized rank

$$u(i) = \frac{\mathbf{ord}(i)}{p} \in (0, 1].$$

Define a small oscillatory component

$$f_{\mathbf{smooth}}(i) = 0.25 \sin(6\pi u(i)) + 0.15 \sin(11\pi u(i)).$$

The overall latent regression surface is

$$f_0(i) = f_{\mathbf{region}}(i) + f_{\mathbf{smooth}}(i), \qquad i = 1, \ldots, p.$$

Thus, $f_0$ is primarily piecewise-constant on a tree-aligned partition with modest smooth variation that follows the tree traversal.

**Noisy replicated observations.** At each vertex $i$, generate $n_{\mathbf{rep}} = 4$ independent replicates:

$$Y_{i\ell} = f_0(i) + \varepsilon_{i\ell}, \qquad \varepsilon_{i\ell} \overset{\text{iid}}{\sim} \mathbf{N}(0, \sigma^2), \quad \ell = 1, \ldots, n_{\mathbf{rep}}.$$

Let

$$\bar{Y}_i = \frac{1}{n_{\mathbf{rep}}} \sum_{\ell=1}^{n_{\mathbf{rep}}} Y_{i\ell}$$

denote the per-location average. Then

$$\bar{Y}_i \mid f_0(i) \sim \mathbf{N}\left(f_0(i),\ \frac{\sigma^2}{n_{\mathbf{rep}}}\right),$$

and we denote $\sigma^2_{\mathbf{eff}} = \sigma^2 / n_{\mathbf{rep}}$ for the effective noise variance of the averages.

The generator produces an irregular spatial domain (hole plus bridge), a neighborhood graph on retained grid cells, and a random MST that acts as the latent geometry. The truth $f_0$ is constructed as a sum of (i) a strong piecewise-constant signal on a contiguous partition induced by removing $K - 1$ tree edges and (ii) a low-amplitude oscillation along a DFS ranking on the same tree, followed by replicated Gaussian observations at each node.

*Table 4.* **Simulated tree–aligned traffic flow with bottlenecks with** $\sigma = 0.15$. Root mean squared error (RMSE) for tree-aligned regression on constrained grid domains of increasing size. Lower values indicate better performance.

| Grid size | PR-BAST | BART | BAST | GP |
|---|---|---|---|---|
| $12 \times 12$ | **0.072** | 0.187 | 0.111 | 0.265 |
| $14 \times 14$ | **0.074** | 0.189 | 0.130 | 0.256 |
| $16 \times 16$ | **0.072** | 0.197 | 0.129 | 0.341 |
| $18 \times 18$ | **0.073** | 0.197 | 0.136 | 0.268 |
| $20 \times 20$ | **0.072** | 0.190 | 0.146 | 0.299 |
| $22 \times 22$ | **0.072** | 0.199 | 0.138 | 0.263 |
| $24 \times 24$ | **0.071** | 0.199 | 0.147 | 0.275 |
| $26 \times 26$ | **0.072** | 0.197 | 0.166 | 0.334 |

*Table 5.* **Simulated tree–aligned traffic flow with bottlenecks with** $\sigma = 0.25$. Root mean squared error (RMSE) for tree-aligned regression on constrained grid domains of increasing size. Lower values indicate better performance.

| Grid size | PR-BAST | BART | BAST | GP |
|---|---|---|---|---|
| $12 \times 12$ | **0.115** | 0.188 | 0.165 | 0.287 |
| $14 \times 14$ | **0.117** | 0.190 | 0.166 | 0.274 |
| $16 \times 16$ | **0.116** | 0.197 | 0.163 | 0.357 |
| $18 \times 18$ | **0.116** | 0.197 | 0.158 | 0.277 |
| $20 \times 20$ | **0.115** | 0.190 | 0.174 | 0.309 |
| $22 \times 22$ | **0.114** | 0.200 | 0.163 | 0.272 |
| $24 \times 24$ | **0.114** | 0.199 | 0.179 | 0.283 |
| $26 \times 26$ | **0.114** | 0.197 | 0.191 | 0.343 |

### D.1.2. ADDITIONAL SIMULATION STUDIES

Additional simulation studies with varying values of noise variance $\sigma^2$ under the above set up is presented in Tables 4 and 5.

### D.2. Simulated Tree-aligned Traffic Flow on One-way Road Network with Flyovers

#### D.2.1. ADDITIONAL DETAILS ON THE DATA GENERATING MECHANISM

**Spatial Domain and Grid Construction**

We begin with a rectangular grid of size $n_r \times n_c$, with

$$n_r = 22, \qquad n_c = 28,$$

yielding $p = n_r n_c$ spatial locations. Each location is indexed by a row–column pair $(r, c)$ and embedded in the unit square according to

$$x(r, c) = \frac{c - 1}{n_c - 1}, \qquad y(r, c) = \frac{r - 1}{n_r - 1}.$$

These coordinates are used only for visualization and for methods that rely on Euclidean geometry; they play no role in the generation of the underlying signal.

**Directed Road Network with One-Way Constraints**

A directed graph $G_{\mathbf{dir}} = (\mathcal{X}, E_{\mathbf{dir}})$ is constructed on the grid to encode one-way traffic rules:

- *Horizontal movement:* In the left half of the grid ($c \leq \lfloor n_c/2 \rfloor$), edges are directed eastward, while in the right half ($c > \lfloor n_c/2 \rfloor$), edges are directed westward.

- *Vertical movement:* All vertical edges are directed southward.

Each directed edge represents a feasible road segment. This construction induces strong anisotropy and non-reversibility in the network, so that Euclidean proximity does not coincide with graph connectivity.

### Flyover Construction

To model highway flyovers or overpasses, we introduce a set of long-range connections that are spatially distant but graph-short. Specifically, we select $K_{\mathbf{fly}} = 34$ disjoint pairs of grid locations that are far apart in Euclidean distance, using a greedy farthest-pair selection scheme.

For each selected pair $(i, j)$, we add a *bidirectional flyover* by inserting two directed edges $i \to j$ and $j \to i$ into $G_{\mathbf{dir}}$. These flyovers substantially reduce graph distances while preserving large Euclidean separation, creating a deliberate mismatch between geometric and network notions of locality.

### Undirected Graph and Spanning Tree Geometry

From the directed graph, we form an undirected graph $G_{\mathbf{und}}$ by ignoring edge directions while retaining edge types (road versus flyover). To define an intrinsic geometry on the domain, we construct a minimum spanning tree (MST) $T_0$ on $G_{\mathbf{und}}$ using edge weights

$$w(e) = \begin{cases} 0.03, & \text{if } e \text{ is a flyover,} \\ 1.0, & \text{if } e \text{ is a road.} \end{cases}$$

This choice ensures that flyovers are preferentially included in the spanning tree. The resulting tree

$$T_0 = (\mathcal{X}, E_{T_0})$$

is connected and acyclic, and serves as the structural backbone for the data-generating process. Distances and neighborhoods used in subsequent steps are defined exclusively with respect to this tree.

### Latent Traffic Intensity Field

Let $L_0$ denote the (unnormalized) graph Laplacian of the spanning tree $T_0$. We generate a latent field $\boldsymbol{\beta}_0 \in \mathbb{R}^p$ from a Gaussian distribution with precision matrix

$$Q_0 = \lambda_0 L_0 + \eta_0 I_p,$$

where $\lambda_0 = 30$ controls the degree of smoothness along the tree and $\eta_0 = 10^{-2}$ is a small ridge term for numerical stability. This construction induces strong dependence between nodes that are close in the tree metric, regardless of their Euclidean separation.

### Tree-Based Smoothing Operator

To obtain the regression signal, the latent field $\boldsymbol{\beta}_0$ is smoothed using a tree-localized kernel. For each node $i$, weights are assigned to nodes $j$ according to

$$W_{ij} \propto \exp\left(-\tau_0 \, d_{T_0}(i, j)^2\right), \qquad \tau_0 = 0.55,$$

where $d_{T_0}(i, j)$ denotes graph distance on the tree. The weights are normalized so that $\sum_j W_{ij} = 1$, and computation is restricted to nodes within a fixed graph-radius neighborhood to ensure sparsity.

The true regression function is then defined as

$$\mathbf{f}_0 = W_0 \boldsymbol{\beta}_0,$$

where $W_0$ is the resulting row-stochastic smoothing matrix. Finally, $\mathbf{f}_0$ is linearly rescaled to ensure strictly positive traffic intensity values.

### Observed Data

The observed response at each location is generated according to

$$y_i = f_0(x_i) + \varepsilon_i, \qquad \varepsilon_i \overset{\mathbf{iid}}{\sim} \mathcal{N}(0, \sigma^2).$$

This completes the data-generating mechanism. The resulting signal is intrinsically aligned with the spanning-tree geometry and incorporates both local smoothness and long-range dependencies induced by flyovers, making Euclidean-based regression methods inherently misspecified.

*Table 6.* **Simulated tree-aligned traffic flow on one-way road networks with flyovers** ($\sigma = 0.1$). Accuracy measured by RMSE for traffic flow estimation on one-way road networks with flyovers. Lower values indicate better performance.

| Grid size | BART | BAST | GP | PR-BAST |
|---|---|---|---|---|
| $12 \times 15$ | 0.572 | 0.230 | 0.545 | **0.074** |
| $14 \times 18$ | 0.577 | 0.299 | 0.577 | **0.075** |
| $16 \times 20$ | 0.540 | 0.257 | 0.546 | **0.078** |
| $18 \times 23$ | 0.450 | 0.294 | 0.502 | **0.065** |
| $20 \times 25$ | 0.442 | 0.393 | 0.555 | **0.052** |
| $22 \times 28$ | 0.401 | 0.381 | 0.521 | **0.070** |
| $24 \times 31$ | 0.361 | 0.385 | 0.528 | **0.056** |
| $26 \times 33$ | 0.385 | 0.459 | 0.532 | **0.067** |

*Table 7.* **Simulated tree-aligned traffic flow on one-way road networks with flyovers** ($\sigma = 0.25$). Accuracy measured by RMSE for traffic flow estimation on one-way road networks with flyovers. Lower values indicate better performance.

| Grid size | BART | BAST | GP | PR-BAST |
|---|---|---|---|---|
| $12 \times 15$ | 0.569 | 0.339 | 0.545 | **0.127** |
| $14 \times 18$ | 0.576 | 0.386 | 0.577 | **0.125** |
| $16 \times 20$ | 0.538 | 0.319 | 0.546 | **0.130** |
| $18 \times 23$ | 0.445 | 0.349 | 0.502 | **0.113** |
| $20 \times 25$ | 0.438 | 0.435 | 0.554 | **0.095** |
| $22 \times 28$ | 0.401 | 0.419 | 0.522 | **0.120** |
| $24 \times 31$ | 0.371 | 0.413 | 0.529 | **0.097** |
| $26 \times 33$ | 0.389 | 0.472 | 0.533 | **0.114** |

### D.2.2. ADDITIONAL SIMULATION STUDIES

Additional simulation studies with varying values of noise variance $\sigma^2$ under the above set up is presented in Tables 6 and 7.

---

**Algorithm 1** PR-BAST posterior computation (Gaussian model, Laplacian prior)

---

**Input:** Observations $\{(v_i, y_i)\}_{i=1}^n$ on $G = (V, E)$ with $p = |V|$; number of components $M$; hyperparameters $(a_\sigma, b_\sigma)$, $(a_\lambda, b_\lambda)$, $(a_\tau, b_\tau)$; regularization $\eta > 0$; number of iterations $S$.

**Output:** Posterior draws $\{T_m^{(s)}, \tau_m^{(s)}, \lambda_m^{(s)}, \boldsymbol{\beta}_m^{(s)}, \sigma^{2(s)}\}_{m=1}^M$ for $s = 1, \ldots, S$.

**Initialize:** For each $m = 1, \ldots, M$, draw $T_m^{(0)} \sim p(T)$, set $\tau_m^{(0)}, \lambda_m^{(0)}$, form $W_m^{(0)} = W_m(\tau_m^{(0)}, T_m^{(0)})$, and initialize $\boldsymbol{\beta}_m^{(0)}$. Initialize $\sigma^{2(0)}$.

**for** $s = 1, \ldots, S$ **do**

    **for** $m = 1, \ldots, M$ **do**

        **Partial residual:** $\mathbf{r}_m \leftarrow \mathbf{y} - \sum_{\ell \neq m} W_\ell^{(s-1)} \boldsymbol{\beta}_\ell^{(s-1)}$.

        **(1) Update** $\boldsymbol{\beta}_m$ **(Gaussian):** Set

$$P_m \leftarrow \frac{1}{\sigma^{2(s-1)}}(W_m^{(s-1)})^\top W_m^{(s-1)} + \lambda_m^{(s-1)} L_{T_m^{(s-1)}} + \eta I_n, \quad b_m \leftarrow \frac{1}{\sigma^{2(s-1)}}(W_m^{(s-1)})^\top \mathbf{r}_m.$$

      Solve $P_m \mu_m = b_m$ and sample $\boldsymbol{\beta}_m^{(s)} \sim \mathrm{N}(\mu_m, P_m^{-1})$ using sparse Cholesky.

      **(2) Update** $\lambda_m$ **(MH on** $\log \lambda_m$**):** Propose $\log \lambda_m' = \log \lambda_m^{(s-1)} + \xi, \xi \sim \mathrm{N}(0, s_\lambda^2)$. Accept with probability

$$\alpha_\lambda = \min\{1, \exp(\Delta_\lambda)\},$$

      where

$$\Delta_\lambda = \tfrac{1}{2} \log \frac{|Q_{T_m}(\lambda_m')|}{|Q_{T_m}(\lambda_m^{(s-1)})|} - \tfrac{1}{2}(\boldsymbol{\beta}_m^{(s)})^\top (Q_{T_m}(\lambda_m') - Q_{T_m}(\lambda_m^{(s-1)}))\boldsymbol{\beta}_m^{(s)} + (a_\lambda - 1) \log \frac{\lambda_m'}{\lambda_m^{(s-1)}} - b_\lambda(\lambda_m' - \lambda_m^{(s-1)}),$$

      with $Q_T(\lambda) = \lambda L_T + \eta I_n$.

      **(3) Update** $\tau_m$ **(MH on** $\log \tau_m$**):** Propose $\log \tau_m' = \log \tau_m^{(s-1)} + \zeta, \zeta \sim \mathrm{N}(0, s_\tau^2)$. Form $W_m' = W_m(\tau_m', T_m^{(s-1)})$ and compute

$$\Delta_\tau = -\frac{1}{2\sigma^{2(s-1)}}\left(\|\mathbf{r}_m - W_m'\boldsymbol{\beta}_m^{(s)}\|_2^2 - \|\mathbf{r}_m - W_m^{(s-1)}\boldsymbol{\beta}_m^{(s)}\|_2^2\right) + (a_\tau - 1)\log\frac{\tau_m'}{\tau_m^{(s-1)}} - b_\tau(\tau_m' - \tau_m^{(s-1)}).$$

      Accept with probability $\alpha_\tau = \min\{1, \exp(\Delta_\tau)\}$.

      **(4) Update spanning tree** $T_m$ **(edge-swap MH):** Propose $T_m'$ by adding $e \in E \setminus T_m^{(s-1)}$, forming a unique cycle, and deleting a uniformly chosen edge on the cycle. Form $W_m' = W_m(\tau_m^{(s)}, T_m')$ and $L_{T_m'}$. Accept with probability $\alpha_T = \min\{1, \exp(\Delta_T)\}$, where

$$\Delta_T = -\frac{1}{2\sigma^{2(s-1)}}\left(\|\mathbf{r}_m - W_m'\boldsymbol{\beta}_m^{(s)}\|_2^2 - \|\mathbf{r}_m - W_m^{(s)}\boldsymbol{\beta}_m^{(s)}\|_2^2\right) - \tfrac{1}{2}(\boldsymbol{\beta}_m^{(s)})^\top \lambda_m^{(s)}(L_{T_m'} - L_{T_m^{(s-1)}})\boldsymbol{\beta}_m^{(s)} +$$

$$\tfrac{1}{2}\left(\log|Q_{T_m'}(\lambda_m^{(s)})| - \log|Q_{T_m^{(s-1)}}(\lambda_m^{(s)})|\right) + \log\frac{p(T_m')}{p(T_m^{(s-1)})}.$$

**(5) Update** $\sigma^2$ **(conjugate):** Let $\mathbf{e} \leftarrow \mathbf{y} - \sum_{m=1}^M W_m^{(s)}\boldsymbol{\beta}_m^{(s)}$ and sample

$$\sigma^{2(s)} \sim \mathbf{Inv\text{-}Gamma}\left(a_\sigma + \frac{n}{2}, \; b_\sigma + \frac{1}{2}\|\mathbf{e}\|_2^2\right).$$

---

