# OpenReview forum: "Probabilistically-routed Bayesian Additive Spanning Trees for Learning on Constrained Domains"
_ICML.cc/2026/Conference — ICML 2026 regular_

### Official Review · Reviewer_1Z32 · 2026-03-11

**Soundness:** 4
**Presentation:** 4
**Significance:** 3
**Originality:** 3
**Overall Recommendation:** 5
**Confidence:** 3

**Summary:**

The authors propose PR-BAST, a natural extension of BAST and earlier BART that aim to overcome the shortcomings of its two predecessors.

The paper starts by giving a compact overview of the framework of non-parametric regression problem: finding the unknown function that describes the relationship between the input and output on a constrained and highly complicated domain, with a centered Gaussian noise whose variance is also unknown. BART (Regression Tree) was invented to provide a specific formulation of the function f as a summation of sub-functions that take a collections of binary decision tree and leaf parameters as the input. Later, BAST (Spanning Tree) was built upon BART that replaces the regression trees with spanning trees, increasing its versatility and performance.

However, both BAST and BART are limited by the deterministic partitions. Therefore, PR-BAST is created such that the spanning trees are created using random partitions. The paper then introduces the setup and the definition of weights for each probable spanning trees, then its asymptotic properties, induced Gaussian random fields and its theoretical advantages.

The paper concludes with experiments on 3 distinct, real-life and/or real-life inspired settings and compares the results with BAST, BART and a Gaussian Process with isotropic Euclidean RBF kernel.

**Compliance With Llm Reviewing Policy:**

Affirmed.

**Final Justification:**

The authors have addressed my concerns so I raised the score. Looking forward to see this paper in the venue.

**Key Questions For Authors:**

I am happy to give the paper a score of accept but I would still encourage the authors to address my questions to make the paper even better.

1. How can you refactor the paper so that a reader with generic background can easily understand the paper, without referring to the original papers for BART and BAST? Can you tune down the technicality and add intuitive illustrations? The paper already puts the full algorithm in appendix, which is a sacrifice for readability. It seems that to achieve full reader-friendliness, the authors need to do more, so that the action of putting the full algorithm in appendix will not go in vain. Meanwhile, refine the appendix that the bridges between the main paper and the appendix is smooth and self-contained. (Like adding a notation table and a preliminary sections for basic math)

2. How can you justify that the algorithm is still widely applicable given all the 8 assumptions. Which of the assumptions can be added without losing generality? Which of the assumptions can be added without reducing the scope of the algorithm too much. Can you quantify the scope loss (for example, adding assumption X only prohibits the algorithm from working on some extreme edge cases, etc.)?

3. Can you address the ambiguity problems I brought up in the previous section and overall refine the paper?

**Limitations:**

Yes.

**Strengths And Weaknesses:**

Strength: Compelling narratives and clear motivations. Solid mathematical foundations and detailed, convincing experiments results. Moreover, the appendix provides a solid breakdown of all the deductions and math in the main paper, as well as a detailed algorithm chart for the main algorithm (which is very complex), completing the overall package.

Weakness: My main feedback in this regard is with the presentation of the paper.

While solid, the paper is not easy to read and as someone who is not directly familiar with the prior work like BART and BAST, I find myself reading the prior papers to fully understand the details. In particular, the paper of BAST offers illustrations of the process (Zhao Tao Luo et al, 2021 has a good illustration in figure 1) and since PR-BAST is more complicated (adding probabilistic components to replace deterministic one), it is hard to digest without proper introduction and illustrations.

Moreover, since the paper is dense, the authors should spend effort making it easy to reader for non-experts on the topic. The writing can be refined, especially during the setup phase. For example, the term "Riemannian Manifold" is only brought up once. I understand that this is the most general setting, but specific properties of Riemannian manifolds are not used. The paper just requires the space to be a generic metric space. Say that first, then probably proceed to say that the most rigorous and general settings are Riemannian Manifold. This will reduce the confusion for readers who are not familiar with Riemannian Geometry. Another example is in section 2.1, the authors wrote "Such partitions provide a natural mechanism for representing spatial heterogeneity while respecting the geometry encoded
by the underlying graph". So the authors mean that not all partitions are "valid" and it has to partition the space in a certain way that respects the underlying structure. So if we are to be mathematical, "natural" does not mean "canonical", meaning there will be ambiguity as to whether a partition represents the "spatial heterogeneity". This needs to be stated clearly here.

Another specific places I want to point out is the 8 assumptions the author explains why they are necessary but I am not entirely clear on why given all the 8 assumptions (which is a lot), the algorithm is still general enough and can be applied to a wide variety of problems. For example, assumption 4.8 says each T_m is a spanning tree of G with uniformly bounded degree. What is the degree is not uniformly bounded? Can that happen in a wide variety of cases? Or can it be reduced to the case where the bound is uniform (so we can add the assumption without losing generality)?

---

> ### Author Rebuttal · Authors · 2026-03-29
>
> We thank the reviewer for this thoughtful question regarding the assumptions. We would like to emphasize that these assumptions primarily serve a technical role in enabling a clear theoretical comparison between hard partitioning (BAST) and probabilistic routing (PR-BAST). In particular, the main theoretical result for PR-BAST relies on four assumptions, each of which has a natural interpretation in graph-based learning settings. The smoothness condition $f_0 \in \mathcal{F}_{\mathbf{sm}}(C)$ formalizes a standard notion of regularity for graph-indexed signals and controls approximation bias. The bounded-degree condition on spanning trees excludes pathological graph constructions while being satisfied by common graphs such as kNN graphs, lattices, and network structures. The prior condition on the temperature parameter $\tau_m$ ensures that the model avoids degenerate regimes with excessively rough behavior, and the well-conditioning of the Laplacian precision guarantees numerical and statistical stability of the Gaussian field representation.
>
> Importantly, these assumptions do not impose restrictive structural constraints on typical applications, but rather exclude extreme or ill-posed cases. In particular, unlike BAST, the PR-BAST result does not depend on any analogue of the cut-complexity condition $g_n^\star$, which is the primary source of limitation under smooth truth. Instead, the assumptions align with standard regularity conditions used in graph-based nonparametric modeling, and therefore do not materially reduce the practical scope of the method. While some of these conditions could be relaxed with additional technical work, doing so would significantly increase complexity without significantly altering the main insights.
>
> To further improve accessibility, we also plan to use the additional page in the camera-ready version to provide a gentler introduction to BART and BAST, so that readers less familiar with this literature can better understand the assumptions and the resulting advantages of PR-BAST.
>
> We hope these clarifications address your key questions..

---

> > ### Author Rebuttal · Reviewer_1Z32 · 2026-04-01
> >
> > Thank the authors for the responses. Looking forward to see this paper in the venue.

---

### Official Review · Reviewer_4Hw3 · 2026-03-13

**Soundness:** 3
**Presentation:** 2
**Significance:** 2
**Originality:** 3
**Overall Recommendation:** 4
**Confidence:** 1

**Summary:**

This submission introduces Probabilistically routed Bayesian Additive Spanning Trees (PR‑BAST), a modification of BAST intended for regression on graph or constrained domains where signals may be smooth along connectivity rather than piecewise constant. The key modeling contribution is to replace hard edge cuts on spanning trees with soft routing weights along a spanning tree.

**Compliance With Llm Reviewing Policy:**

Affirmed.

**Final Justification:**

The authors addressed my comments, and I am comfortable maintaining my score.

**Key Questions For Authors:**

1. What exact distribution over spanning trees is used in experiments and theory? If MST with iid random weights is used, please characterize it; if uniform is intended, how is it sampled?

2. Can you add a graph aware GP/GRF baseline (e.g., Laplacian kernel, diffusion kernel, intrinsic graph distance kernel)? Given the GRF connection, this seems essential.

3. How sensitive are results to M, \tau prior, \lambda prior, and truncation radius r?

4. In the one‑way road network simulation, you construct a directed graph then ignore direction to form an undirected graph for the MST. Does PR‑BAST handle directionality at all, and would direction aware distances matter?

**Limitations:**

Yes

**Strengths And Weaknesses:**

Strengths:

1. The gap between piecewise constant tree partition models and smooth graph signals is discussed well and introducing a soft routing operator is a natural way to trade off locality continuously using \tau.

2. Using a tree Laplacian precision makes sparse Cholesky feasible, and the MCMC steps explicitly leverage sparsity, plus optional truncation of routing neighborhoods for sparse W rows.

3.  In the synthetic domains where Euclidean proximity is misleading, PR‑BAST looks well‑matched to the data generator and yields substantially lower RMSE than BAST/GP/BART in the reported tables; the NYC taxi experiment similarly reports the best RMSE and sMAPE among the listed methods.

Weaknesses:

1. The paper describes generating T_m via random edge weights and a minimum spanning tree construction, then separately states a default uniform prior over spanning trees. An MST under iid continuous edge weights does not generally yield a uniform distribution over spanning trees. If the intended prior is MST‑induced, that should be stated as the actual prior and its implications discussed; if uniform is intended, the MST mechanism is not correct for sampling uniformly. This matters for both theory assumptions and practical mixing.

2. The GP baseline in experiments is described as using an isotropic Euclidean RBF kernel, which is known to fail in non‑Euclidean graph geometries. A more fair comparator would include a graph GP / GRF baseline using Laplacian‑based kernels or intrinsic graph distances (especially since the proposed method is closely related to Laplacian regularization through LTL_T). Without stronger baselines, the empirical analysis is lacking.

3. Several implementation details that affect reproducibility and credibility are not fully specified in the main text. For instance, how M is chosen, hyperparameter settings for Gamma/Inv‑Gamma priors, truncation radius rules for W etc,.

4. The paper repeatedly emphasizes retaining interpretability of tree‑based models, but PR‑BAST components are now dense weighted averages across nodes along a tree, and the learned object is an ensemble of spanning trees + temperature‑controlled kernels. This may still be interpretable in terms of routing geometry, but it is not the same interpretability as cut‑based partitions with constant regions. A better interpretation could help.
5. Grammar issues and notational inconsistencies should be addressed to improve readability.

---

> ### Author Rebuttal · Authors · 2026-03-29
>
> We thank the reviewer for the careful reading and constructive feedback.
>
> Regarding the spanning tree distribution, we agree that this point should be clarified more explicitly. In our current implementation, spanning trees are generated by assigning i.i.d.\ random edge weights and computing the corresponding minimum spanning tree, which induces a valid randomized distribution over spanning trees. Our intention was to use this as a computationally convenient mechanism for sampling geometry-respecting trees, rather than to approximate a uniform spanning tree distribution. We agree that this distinction is important, and will revise the manuscript to clearly state the induced distribution and discuss its implications for both theory and practice.
>
> We also agree that including a graph-aware GP/GRF baseline would strengthen the empirical evaluation, particularly given the connection between each spanning tree of PR-BAST and graph-based Gaussian random fields.
> We note that while graph-aware GP substantially improves over Euclidean GP by incorporating connectivity, it still falls short of PR-BAST. This is because PR-BAST combines geometry-aware smoothing with adaptive, multi-scale representation through its ensemble of spanning trees, allowing it to capture both global smooth variation and localized structural heterogeneity more effectively than a single-kernel GP/GRF model.
> Results from experiments under the setup in Section 5.1 are presented as support to our claim:
>
> Grid: BART / SoftBART / BAST / Graph-Aware GP / GP / PR-BAST
>
> $12\times12$: 0.21 / 0.20 / 0.19 / 0.20 / 0.31 / 0.15
>
> $14\times14$: 0.21 / 0.20 / 0.19 / 0.20 / 0.29 / 0.16
>
> $16\times16$: 0.20 / 0.20 / 0.20 / 0.21 / 0.37 / 0.16
>
> $18\times18$: 0.19 / 0.20 / 0.20 / 0.21 / 0.29 / 0.16
>
> $20\times20$: 0.20 / 0.20 / 0.19 / 0.20 / 0.32 / 0.16
>
> $22\times22$: 0.19 / 0.20 / 0.20 / 0.21 / 0.28 / 0.15
>
> $24\times24$: 0.20 / 0.20 / 0.20 / 0.21 / 0.29 / 0.15
>
> $26\times26$: 0.22 / 0.21 / 0.20 / 0.21 / 0.35 / 0.15
>
> Results on other simulations and real data are show similar pattern, and is skipped here due to space constraints.
>
> Regarding implementation details: In the revision, we will provide explicit details on the choice of $M$, prior specifications for $\tau$ and $\lambda$, and the truncation radius $r$, along with practical guidelines used in our experiments. We will also include sensitivity analyses to demonstrate robustness across a reasonable range of these parameters.
>
> On interpretability, we appreciate this observation. While PR-BAST differs from the piecewise-constant partition interpretation of BAST, interpretability is retained in terms of routing geometry along spanning trees, with temperature controlling the locality of influence. We will expand this discussion to better articulate how interpretability manifests in the softened setting.
>
> Regarding directionality, we thank the reviewer for highlighting this important point. In the current formulation, PR-BAST operates on an undirected representative graph to maintain simplicity in exposition. That said, we agree that in settings where directional effects are intrinsic (e.g., flow dynamics, travel times, or asymmetric interactions), incorporating direction-aware distances could be important. The PR-BAST framework can naturally accommodate this by replacing the underlying distance metric with one derived from directed graphs (e.g., shortest path distances on directed edges or asymmetric Laplacians), which would lead to direction-sensitive routing weights.
>
> Finally, we will carefully revise the manuscript to address grammatical issues, improve notation, and enhance overall readability. We also plan to use the additional page in the camera-ready version to provide a gentler introduction to BART and BAST, to improve accessibility for readers less familiar with the literature.
>
> We hope these clarifications address the reviewer’s concerns and improve the presentation and empirical strength of the paper.

---

> > ### Author Rebuttal · Reviewer_4Hw3 · 2026-04-03
> >
> > I don't have any further comments; I thank the authors for their responses.

---

### Official Review · Reviewer_hRzD · 2026-03-16

**Soundness:** 3
**Presentation:** 3
**Significance:** 2
**Originality:** 2
**Overall Recommendation:** 4
**Confidence:** 2

**Summary:**

The paper proposes *probabilistic-routed Bayesian additive spanning trees* (PR-BAST), which relaxes simpler BAST models by allowing a *soft / probabilistic* version of the assignments from the spanning-tree model components. The motivation is to obtain a more suitable representation of smoothly varying functions on constrained domains. The key model extension is achieved by replacing tree-conditional piecewise-constant mappings with weighted averages over tree leaves, where the weights are computed from pairwise (non-Euclidean) distances between points under a spanning tree consistent with a (pre-specified) graph representing the geometry. The paper presents theoretical advantages of PR-BAST over BAST for certain data-generating processes, and it shows empirical gains in synthetic experiments.

**Compliance With Llm Reviewing Policy:**

Affirmed.

**Final Justification:**

The rebuttal addressed my main concerns, so I decided to increase my score by one rank.

**Key Questions For Authors:**

1. Could the authors clarify whether $G$ is treated as *fixed and fully known*, whether one could place a prior on $G$ or learn it from data, and what failure modes arise when $G$ is poorly specified, not representative of $\mathcal{M}$ or very dense?
2. Since the main methodological step seems closely related in spirit to the move from BART to softBART, could the authors clarify more explicitly how PR-BAST differs conceptually and technically from applying soft/probabilistic tree-routing ideas in the BAST setting? Also, is there a reason why the experiments compare mainly against BART rather than including softBART as a more natural baseline?  Will the authors rephrase the contribution of the paper in light of the priorly made contrast of BART vs softBART?
3. The paper is motivated by regression on non-Euclidean domains, but in the final construction the geometry seems to enter only through the representative graph $G$ and then through the tree-based distance $d_T$. Could the authors discuss more explicitly what geometric structure from the original Riemannian manifold $\mathcal{M}$ is preserved, what is discarded, and under what conditions this reduction is expected to be adequate?
4. I am not sure I fully understand the framing of Assumption 4.1. Why should the **true** $|f_0(x)|$ be uniformly bounded by the log-size of the dataset $\log n$. The function $f_0$ is real-valued (1D) and its bounds should have nothing to do with the number of samples. Wouldn't it be sufficient to say $|f_0(x)|$ is uniformly bounded by a constant (not depending on $n$)? Am I misunderstanding a key rationale?
5. Is the **prediction** task graph-agnostic? Only the new location $v_*$ is needed, without needed a new graph $G_*$ on $\mathcal{X}=(x\_1,x\_2,...,x\_n,v\_* )$? Notice that if $G$ was computed from $\mathcal{M}$, then a new $G_*$ could be computed from the same Riemmanian geometry including the new point $v_*$, so a more graph-based prediction could also be done.

**Limitations:**

I agree with the authors that the potential societal consequences of this work are fuzzy and so they don't have to be highlighted within the paper.

**Strengths And Weaknesses:**

**Soundness**: The paper is technically sound, and its claims are generally well supported by the theoretical insights provided. While some important technical aspects are left relatively undiscussed, such as the properties and requirements of the Riemannian manifold $\mathcal{M}$ and its representative measured graph $G$, the reasoning and deductions that are presented appear mathematically rigorous and internally consistent.

**Presentation**:   The paper is well written, and it develops its main ideas and derivations in a clear order and with good detail.

There are, however, some omissions that make the reading flow and overall understanding less than ideal for a curious reader. For instance:
- The paper only briefly mentions the role of the graph $G$, but it does not discuss in enough detail its origin, specification, and potential failure modes, despite its central importance for the proposed methodology. The existence and use of $G$ is precisely what motivates (PR-)BAST over simpler BART models, so I think a fuller discussion of how $G$ is constructed and used would be beneficial. Is $G$ treated as *fixed and given*? What happens when $G$ is very dense, so that the number of spanning trees grows super-exponentially? Is it possible to construct, learn, or place a prior on $G$?
- Some technical claims and assumptions could be stated more rigorously. For instance, Assumption 4.1 says:
> The true signal satisfies $\max\_i |f_0(s_i)|\leq C_0\log n$.

but $f_0$, $C_0$ and $s_i$ have not been properly introduced beforehand. A more explicit phrasing would be
> Let $f_0$ be the true signal and let $n=|\mathcal{X}|$. Then there exists a constant $0<C_0<\infty$, such that for all $x\in\mathcal{X}$, $f_0(x)$ is uniformly bounded by $n$ as $|f_0(x)|\leq C_0\log n$.

(See also my further questions below regarding this assumption.)

- Other textual and technical parts could also benefit from greater explicitness. For instance, does the method assume a mean-centered outcome $y$, or do the variables $x$ include a constant unit component in $\mathcal{M}$?

**Significance**: The paper addresses an interesting problem, namely extending BAST models to allow for more appropriate modeling of genuinely smooth functions with inputs in non-Euclidean spaces. The scope strikes a good balance between being general and being specialized. I can also see interesting extensions, such as introducing a link function for categorical outcomes, or even modeling outcomes that themselves lie on Riemannian manifolds, that is, $y\in\mathcal{M}_Y$ rather than being real-valued.

Still, there are a couple of concerns that somewhat weaken the significance.
- The paper emphasizes, in a few places including the _Our contribution_ paragraph, that tree-based baselines retain interpretability. While this is certainly true for individual tree models, BART and BAST are tree-ensemble models, and it is not automatic to describe them as "interpretable" without a more careful discussion of what exactly is meant by that term. One could argue that BART/BAST models are less interpretable than a single decision tree, but more interpretable than other tree ensembles such as random forests because, among other reasons, they provide posterior inclusion probabilities that directly measure feature importance and thus aid interpretation. This nuance is not sufficiently addressed in the paper.
- I believe the non-Euclidean geometry plays a smaller role in the methodology than one might initially expect. In practice, it is collapsed into a representative graph of geodesic proximity $G$, and although other metrics are said to be admissible, the proposal ultimately represents geometry only through the tree-based distance $d_T$, which may discard a substantial amount of geometric information. Yet the paper does not discuss clearly what is lost or gained when moving from a Riemannian-manifold geometry to a graph- or tree-induced geometry.

**Originality**:  The contributions of the paper are incremental, in the sense that they combine ideas, tools, and methods that already exist in the literature. For instance, both BAST and softBART can be viewed as generalizations of BART, the former to accommodate non-Euclidean inputs and the latter to allow probabilistic leaf rules. The key idea of this paper is to bring these two motivations and solutions together. I do think the proposed approach is more than a simple juxtaposition of existing components, and I congratulate the authors on the creativity required to make these ideas work jointly. Still, I have some critical comments regarding how the contribution is presented:
- While the paper cites *Linero & Yang (2018)*, which introduced softBART as a flexibilization of BART in which tree decisions are treated probabilistically, it does not clearly acknowledge that contribution in the framing of the paper. Without such a discussion, an unfamiliar reader might come away with the impression that softening indicator functions in tree-ensemble models is itself a major contribution here. A more transparent presentation would explain that the motivation for softening leaf rules in BAST is analogous to the earlier move from BART to softBART. In fact, it is my understanding that the experiments use BART as a benchmark, rather than the more natural comparison to softBART.

---

> ### Author Rebuttal · Authors · 2026-03-28
>
> **We thank the reviewer for the careful reading of the manuscript.**
>
>  1. **Nature of $G$:**
> In our current formulation, the graph $G$ is treated as fixed, serving as a representation of domain geometry (e.g., spatial adjacency or network structure), from which spanning trees are drawn. The framework naturally admits extensions where $G$ is unknown and one learns $G$ from data. Regarding failure modes, if $G$ is mis-specified or overly dense, the induced tree distances can blur meaningful locality and lead to over-smoothing; if $G$ is not representative of the underlying geometry, the model may fail to capture the true dependency structure. We will add a discussion explicitly outlining these considerations.
>
> 2. **Soft thresholding/probabilistic routing:**
>
> We fully agree that replacing hard functions with continuous relaxations is a broadly influential idea in Statistics and ML, including hard to soft-thresholding in shrinkage estimation, continuous relaxations of combinatorial procedures, softened priors for convenient Bayesian computation, BART to Soft BART,  etc. Our work can be viewed within this broader paradigm. However, the contribution of PR-BAST is not simply the introduction of soft routing, but also how this softening is integrated with domain geometry and structured priors. We shall include a more explicit discussion of the matter in the camera-ready version of the paper.
>
> While both approaches replace hard functions with probabilistic routing, the mechanisms and implications differ fundamentally. In softBART, routing is defined via axis-aligned splits in Euclidean covariate space, with locality governed by coordinate thresholds and bandwidth parameters. In contrast, PR-BAST defines routing through graph geodesic distances on spanning trees, so that locality is determined by connectivity and path distance, which may differ substantially from Euclidean proximity in constrained or non-Euclidean domains. This distinction is not merely representational: conditional on a tree, each component in the PR-BAST ensemble induces a Gaussian random field with Laplacian precision, providing a principled link to graph-based smoothing. More broadly, PR-BAST explicitly incorporates structural information through the underlying graph, whereas softBART operates purely in ambient covariate space without encoding adjacency or connectivity constraints.
>
> We agree that including softBART as an empirical baseline would further strengthen the comparison. While softBART shares the use of continuous, probabilistic routing, it is not a direct competitor to PR-BAST, as it does not incorporate underlying graph structure. Additional results from experiments under the setup in Section 5.1 are presented as support to our claim:
>
> Grid:BART / SoftBART / BAST / GP / PR-BAST
>
> $12\times12$: 0.21 / 0.20 / 0.19 / 0.31 / 0.15
>
> $14\times14$: 0.21 / 0.20 / 0.19 / 0.29 / 0.16
>
> $16\times16$: 0.20 / 0.20 / 0.20 / 0.37 / 0.16
>
> $18\times18$: 0.19 / 0.20 / 0.20 / 0.29 / 0.16
>
> $20\times20$: 0.20 / 0.20 / 0.19 / 0.32 / 0.16
>
> $22\times22$: 0.19 / 0.20 / 0.20 / 0.28 / 0.15
>
> $24\times24$: 0.20 / 0.20 / 0.20 / 0.29 / 0.15
>
> $26\times26$: 0.22 / 0.21 / 0.20 / 0.35 / 0.15
>
> Results on other simulations and real data show similar pattern, and is not presented here due to space constraints.
>
> 3. **Riemannian geometry:**
>
> The Riemannian geometry enters our model through the representative graph $G$, which serves as an approximation to the underlying manifold. In particular, local connectivity and approximate geodesic distances are preserved via the graph and subsequently through tree-based distances, while fine-scale continuous features such as curvature beyond the graph resolution are not explicitly retained. This reduction is appropriate when the graph provides a sufficiently accurate representation of the intrinsic geometry.
>
> 4. **Assumption 4.1:**
>
> The $\log n$ bound is used in the sieve construction in the proof, where we try to control the size of the parameter space and obtain the entropy bound for the covering number. This is a technical device for managing complexity and prior tails, not a modeling restriction. In particular, any uniformly bounded function satisfies this condition for sufficiently large $n$, and it does not limit practical applicability.
>
> 5. **Prediction:**
>
> For prediction at a new location $v_\star$, a principled approach requires constructing the graph including $v_\star$ so that distances and connectivity remain consistent with the underlying geometry. In practice, this can be done efficiently by augmenting the existing graph with $v_\star$, and then recomputing the relevant tree-based distances. This yields a fully geometry-consistent, computationally convenient prediction.
>
> **In light of these clarifications, we appeal to the reviewer to kindly reconsider the rating.**

---

> > ### Author Rebuttal · Reviewer_hRzD · 2026-04-01
> >
> > Thank you for the rebuttal. It addressed my concerns. I believe the intended changes will strengthen the paper, including: (i) explicitly presenting all the assumptions related to the exogeneity, representativity, and sufficiency of $G$; (ii) an acknowledgement and fair comparison against Soft BART; and (ii) a more rigorous wording of Assumption 4.1.
> >
> > I have only one remaining question regarding the prediction step.  In the rebuttal the authors say _"... a principled approach requires constructing the graph..."_, but that procedure is not the one contained in the paper (in the paragraph before section 4). Is that correct? Will the paper recommend graph-based or graph-agnostic prediction? Thank you.

---

> > > ### Author Response · Authors · 2026-04-02
> > >
> > > We thank the reviewer for this question regarding the prediction step. The description in the manuscript was concise due to space limitations and because prediction was not central to the subsequent methodological development. We agree, however, that the current presentation can be made more explicit.
> > >
> > > Importantly, the prediction rule in the paper is inherently graph-based. The computation of routing weights $W_m(v_\star,\cdot)$ requires evaluating distances and connectivity relative to the learned tree structures, which implicitly assumes that the new location $v_\star$ is incorporated into the underlying graph $\mathcal{G}$. Thus, augmenting $\mathcal{G}$ with $v_\star$ is not a modification of the method, but an implicit step already required by the current formulation.
> > >
> > > To clarify this, we will revise the manuscript by explicitly describing prediction as a two-step procedure:
> > >
> > > - **Step 1 (Graph augmentation):** Augment the underlying graph $\mathcal{G}$ to include the new location $v_\star$, preserving the same connectivity and distance structure used during training.
> > >
> > > - **Step 2 (Routing weights and prediction):** Compute the routing weights $W_m(v_\star,\cdot)$ using the same tree-based construction, and form predictions as described in the manuscript.
> > >
> > > We hope this clarification resolves the concern and improves the overall clarity of the manuscript, and we would be grateful if you could reconsider their evaluation in light of this revision.

---

### Official Review · Reviewer_rSsN · 2026-03-16

**Soundness:** 3
**Presentation:** 3
**Significance:** 2
**Originality:** 3
**Overall Recommendation:** 4
**Confidence:** 4

**Summary:**

The work revolves around Bayesian estimation of spanning trees. Previous work, on Bayesian additive spanning tree (BAST) is able to learn complex structures (spanning trees), e.g., traffic flow. However, spanning trees represent piecewise constant functions  thus are not fit to represent smoothly varying functions on constrained domains, such as traffic flow in a city. This work cleverly solves this problem with smoothly varying over the space of all spanning trees. The authors further develop this framework and investigate conditioning over trees and describe their scalable posterior. This leads to faster posterior contraction rates compared to BAST.

**Compliance With Llm Reviewing Policy:**

Affirmed.

**Key Questions For Authors:**

while I agree it is a nitpick: perhaps one would want to change the u index to \hat x. Also, equations 2,3 perhaps to omit the _i subscript.

**Limitations:**

yes

**Strengths And Weaknesses:**

The paper is pleasure to read. It is well structured and easy to follow. Its theoretical contribution is solid, and backed up with non-trivial framework that elegantly developed in the supplementary material. The work is technically sound and its experimental validation is solid, while considering its theoretical nature. The smoothing variation of the BAST is original. While different smoothing techniques could have been researched, it seems the authors came up with a framework that is both theoretically appealing and practical.

Perhaps the weakest point of the work is its significance to the ICML community. This work is better fit to statistical community. Its techniques and motivations are natural to the statistics community and I kindly think the authors would have benefit a better evaluation process in such a community. Nevertheless, the machine learning community has roots in statistics and it is a viable venue to consider.

---

> ### Author Rebuttal · Authors · 2026-03-29
>
> We thank the reviewer for their careful reviewing our paper. We respectfully view the scope of the contribution somewhat differently. While the methodology is indeed grounded in statistical principles, we believe the problem setting and proposed solution are closely aligned with the scope of ICML, particularly in learning on structured and non-Euclidean domains.
>
> Many modern machine learning applications involve data supported on graphs, networks, or manifold-like structures. Examples include transportation systems, environmental and epidemiological data, single-cell transcriptomics, social or biological interaction networks, etc., where standard Euclidean modeling assumptions are inadequate. Our approach provides a principled, interpretable and computationally scalable framework for learning in such settings by providing a geometry-aware flexible function estimation methodology. This enables capturing both global smooth variation and localized structural heterogeneity in the underlying function, which is critical in these applications. We believe these aspects make the proposed framework broadly applicable even beyond the motivating examples considered in the article, and relevant to a wide range of modern ML problems involving structured data.
>
> We hope this clarification is satisfactory, and we kindly request to reconsider their scores in view of this.

---

> > ### Author Rebuttal · Reviewer_rSsN · 2026-04-06
> >
> > Thank you for your kind and respectful response. My comment on ICML vs Statistics is based on my decades of experience in the field. I greatly respect your point of view, but I respectfully maintain my opinion on these fields of research and respectively my score.

---

> > > ### Author Response · Authors · 2026-04-06
> > >
> > > We respect your evaluation. Also, thanks for graciously noting that ICML is still a viable venue for our paper in your intial review. We appreciate it.

---

### Decision · Program_Chairs · 2026-04-30

**Decision:**

Accept (regular)

**Comment:**

The paper proposes softening BAST hard tree partitions into probabilistic routing, while keeping BAST related interpretability and relatively scalable Bayesian inference. Theoretical results seem solid (e.g. faster contraction than BAST under relevant conditions). The applications are interesting with graph- network- signal (e.g. traffic, urban data) and include both simulated and real data cases. In absolute terms, improvements in Tables 1-5 look huge (a bit too consistently good sometimes, to be honest, to question the sufficiency of these examples and the amount of tuning and cherry picking). Broader UCI benchmarks would have been appreciated. More importantly (although the conclusions are likely to hold for these huge effect sizes), statistical evaluation is poor (point values of the performance with not even confidence intervals). Relevant to experimental design, pairwise tests or mixed-effect regression models (with your method acting as a baseline) should be chosen to claim significance of the results, as the current evaluation with only general summaries (Tables 1 - 5) is not rigorous in that sense. If the paper is accepted, I expect this to be revised and discussed.

The reviewers find the paper technically sound, well written, and positioned, yet some aspects relative to softBART and graph GP, as well as interpretability claims, and robustness to misspecification, should have been elaborated more clearly. The rebuttal resolves most of these issues. Taking in consideration reviewers' recommendations and good rebuttal, I recommend a weak accept decision for this paper. That said, upon acceptance, I expect substantially improved statistical evaluation of the results to appear in Tables 1-5 to support the significance of the consistent improvement claims. Also, an honest discussion of the limitations and scenarios where PR-BAST is expected to work not that well will be appreciated.